# *Escherichia coli* Enumeration in a Capillary-Driven Microfluidic Chip with SERS

**DOI:** 10.3390/bios12090765

**Published:** 2022-09-17

**Authors:** Üzeyir Dogan, Ferah Sucularlı, Ender Yildirim, Demet Cetin, Zekiye Suludere, Ismail Hakkı Boyaci, Ugur Tamer

**Affiliations:** 1Department of Analytical Chemistry, Faculty of Pharmacy, Düzce University, 81620 Düzce, Türkiye; 2Department of Analytical Chemistry, Faculty of Pharmacy, Gazi University, Etiler, 06330 Ankara, Türkiye; 3Aselsan A.Ş., Radar, Electronic Warfare Systems Business Sector, 06172 Ankara, Türkiye; 4Department of Mechanical Engineering, Faculty of Engineering, Middle East Technical University, Çankaya, 06800 Ankara, Türkiye; 5Department of Mathematics and Science Education, Gazi Faculty of Education, Gazi University, Besevler, 06500 Ankara, Türkiye; 6Department of Biology, Faculty of Science, Gazi University, Besevler, 06500 Ankara, Türkiye; 7Department of Food Engineering, Hacettepe University, Beytepe, 06800 Ankara, Türkiye

**Keywords:** *Escherichia coli*, microfluidic chip, SERS detection, magnetic separation, gold nanorods

## Abstract

Pathogen detection is still a challenging issue for public health, especially in food products. A selective preconcentration step is also necessary if the target pathogen concentration is very low or if the sample volume is limited in the analysis. Plate counting (24–48 h) methods should be replaced by novel biosensor systems as an alternative reliable pathogen detection technique. The usage of a capillary-driven microfluidic chip is an alternative method for pathogen detection, with the combination of surface-enhanced Raman scattering (SERS) measurements. Here, we constructed microchambers with capillary microchannels to provide nanoparticle–pathogen transportation from one chamber to the other. *Escherichia coli* (*E. coli*) was selected as a model pathogen and specific antibody-modified magnetic nanoparticles (MNPs) as a capture probe in a complex milk matrix. MNPs that captured *E. coli* were transferred in a capillary-driven microfluidic chip consisting of four chambers, and 4-aminothiophenol (4-ATP)-labelled gold nanorods (Au NRs) were used as the Raman probe in the capillary-driven microfluidic chip. The MNPs provided immunomagnetic (IMS) separation and preconcentration of analytes from the sample matrix and then, 4-ATP-labelled Au NRs provided an SERS response by forming sandwich immunoassay structures in the last chamber of the capillary-driven microfluidic chip. The developed SERS-based method could detect 10^1^–10^7^ cfu/mL of *E. coli* with the total analysis time of less than 60 min. Selectivity of the developed method was also tested by using *Salmonella enteritidis* (*S. enteritidis*) and *Staphylococcus aureus* (*S. aureus*) as analytes, and very weak signals were observed.

## 1. Introduction

Food contamination is a very important issue for public health, especially in milk and milk products [1,2,3,4]. Fast, simple and reliable alternative techniques are needed to solve this crucial problem. Plate counting (24–48 h) methods should be replaced by novel biosensor systems as an alternative reliable pathogen detection technique, demonstrated by the use of nanoparticles in last few years [5,6,7]. In these developed biosensor systems, different analytical techniques are used, such as surface enhanced Raman scattering (SERS) [8,9,10,11,12,13,14,15], fluorescence [16,17,18,19], enzyme-linked immunosorbent assay (ELISA) [20,21,22], surface plasmon resonance (SPR) [23,24], electrochemical methods [25,26,27] and colorimetric methods [28,29]. SERS is a very good alternative analytical technique due to its high sensitivity, which can even detect a single bacterium. The SERS technique made the detection of many molecules with weak Raman signals possible. This sensitivity enhancement made Raman spectroscopy a commonly used detection technique. The enhancement phenomenon lies in the application of nanostructured metal, such as silver or gold. Since these metal nanoparticles have enriched optical properties, target molecules can be detected in lower detection limits. In addition, nanoparticles have been preferred for labeling in detection techniques and have enabled multiplex analysis to be carried out. Nanoparticles are also promising in the design of sensor devices, such as SERS substrates, which are ultimate tools for sensitive pathogen detection [30,31,32].

A selective preconcentration step is a very important issue if the analyte concentration is very low or the sample volume is limited for the analysis. In order to solve this issue, magnetic nanoparticles (MNPs) are commonly used because of their magnetic property and they are easily modified to a specific analyte. For example, the removal of 88% *Escherichia coli* (*E. coli*) from the sample has been achieved by using magnetic glyco nanoparticles, as shown in the literature [33]. Similarly, antibodies have been used to functionalize specific MNPs to target bacteria in order to preconcentrate *E. coli* from milk samples by Cheng et al. [34]. In addition, the biocompatible and low toxic properties of MNPs have increased the potential of applications in living organisms.

The usage of microfluidic chips has some important advantages, such as less sample consumption, reduction in the number of analytical steps and the reusable property of the microchip, which makes the studies more economical and effective. Most of the microfluidic chips are continuous flow types, in which the sample and other chemicals are mixed through the channels in the chip. However, pump and port connections must be used in this type of chip, which brings some disadvantages, such as reduced reliability due to the use of off-chip components. Although a few continuous flow type-based microfluidic devices have already demonstrated high sensitivities in the determination of biologically relevant organisms, these approaches require a long analysis time and do not include in-field analysis strategies of target pathogens [35,36]. A circular dielectrophoretic microfluidic device was also utilized for pathogen analysis from human blood in less than 1 min [37]. Krafft et. al. used a low-cost disposable PDMS device for the concentration of bacteria from drinking tap water, which combines filtration with the electrodriven flow [38]. However, these techniques require sophisticated technologies or external apparatuses that are hardly compatible with a portable device. Pipette-operated capillary-driven chips are effective means to eliminate these disadvantages without compromising the advantages of miniaturized systems [39,40]. 

Optical-based systems involve enzymatic biosensors, immunosensors, biosensors for bacteria cells, proteins, and antibodies using opto/electronic transduction of the biological process that occurs at the sensing surface. The biological element can recognize a specific analyte, activity or concentration in solution. The detection can be a binding process and immobilization steps of antibodies on sensor surfaces and nonspecific protein binding, for instance, the binding method consists of an antibody, nucleic acid, receptor protein or cell receptor, and a synthetic receptor. Nanomaterial-based biosensors can offer novel techniques that may perhaps meet the current demand for early and rapid diagnosis. In this study, rapid and sensitive enumeration of *E. coli* is achieved in milk samples by using the SERS technique in a capillary-driven microfluidic chip. *E. coli* was preconcentrated by using antibody-modified MNPs. The sandwich immunoassay structures were formed in the chip by interaction with MNP-*E. coli* conjugates with modified gold nanorods (Au NRs), which were also used as SERS labels. The SERS quantification of *E. coli* in a capillary-driven microfluidic chip with the combination of immunomagnetic separation (IMS) in milk samples, to the best our knowledge, has been reported for the first time in this study. In the first part, nanoparticles were synthesized, modified, and optimized to have higher sensitivity. A calibration graph is constructed by using different concentrations of *E. coli* (10^1^–10^7^ cfu/mL). Selectivity of the developed biosensor system specific to *E. coli* was proven in the presence of *Salmonella enteritidis* (*S. enteritidis*) and *Staphylococcus aureus* (*S. aureus*). The developed immunoassay procedure is successfully applied to spiked milk samples as a real sample.

## 2. Experimental Section

### 2.1. Chemicals

Iron (III) chloride (FeCl_3_), iron (II) sulfate heptahaydrate (FeSO_4_·7H_2_O), hydrogen tetrachloroaurate (HAuCl_4_), silver nitrate (AgNO_3_), hexadecyltrimethylammonium bromide (CTAB), 11-mercaptoundecanoic acid (11-MUA), 4-ATP, ethanolamine, N-(3-dimethylaminopropyl)-N′-ethylcarbodiimide hydrochloride (EDC), ethylenediaminetetraacetic acid (EDTA), hydroxylamine hydrochloride, perchloric acid (HClO_4_), and sodium borohydride (NaBH_4_), sodium hydroxide (NaOH), 2-morpholinoethanesulphonic acid monohydrate (MES), and absolute ethanol were obtained from Merck (Darmstadt, Germany). Hydroxysulfosuccinimide sodium salt (NHS) was obtained from Pierce Biotechnology (Bonn, Germany). NaCl, Na_2_HPO_4_, and KH_2_PO_4_ were purchased from J.T. Baker (Deventer, The Netherlands), used as phosphate-buffered saline (PBS). N- Biotin-conjugated rabbit anti-*E. coli* polyclonal antibodies were received from Abcam plc. (Cambridge, UK). Immunopure avidin was received from Pierce Biotechnology (Rockford, IL, USA). In order to prepare the desired concentrations of solutions, the Millipore milliQ water purification system was used to obtain deionized water (18 MΩ cm).

### 2.2. Microorganisms

*E. coli* (ATCC 35218) and *S. enteritidis* (ATCC BAA 1045) were obtained from Hacettepe University Food Research Center Culture Collection, Ankara, Türkiye. *S. aureus* was obtained from Refik Saydam National Type Culture Collection, Ankara, Türkiye. Tryptic soy broth (TSB; Merck, Germany) was used to grow bacteria cultures at 37 °C for 18 h. The obtained cells were collected by centrifugation at 4000 rpm for 10 min. The bacteria samples were washed three times with sterile PBS (0.67 M at pH 7.4) and then the turbidity of cells was adjusted to 0.5 McFarland standard. All bacterial suspensions were adjusted using the McFarland standard, and the numbers of bacteria in the samples were also confirmed by conventional plating methods. The enumeration of *E. coli* was conducted by spreading the proper dilutions of bacteria on eosin methylene blue agar (EMBA). Bacterial counts were calculated after the incubation at 37 °C for 18–24 h.

### 2.3. Instrumentation

A Spectronics, Genesis model single beam UV–Vis spectrophotometer was used to obtain optical absorption spectra by using quartz cuvettes (1 cm light path). SERS spectra were obtained using the DeltaNu Examiner Raman microscope (Deltanu Inc., Laramie, WY, USA) with a 785-nm laser source and CCD detector. A JEOL JEM 1400 (JEOL Ltd., Tokyo, Japan) was used to obtain transmission electron microscope (TEM) images at 80 kV.

### 2.4. Synthesis of Au NRs

First, 4-ATP-modified Au NRs were employed as Raman labels for the quantification of *E. coli*. Briefly, the seed solution was prepared by adding HAuClO_4_ and 0.01 M NaBH_4_ solutions to 0.1 M CTAB solution, respectively. Then, 0.01 M HAuClO_4_, 0.01 M AgNO_3_, 0.1 M ascorbic acid and seed solutions were added to 0.1 M CTAB solution, respectively. After waiting 2 h for Au NR formation, they were precipitated by centrifugation, washed with ethanol and kept in ethanol for further modification processes. 

### 2.5. Synthesis of Fe_3_O_4_@Au MNPs

The synthesis and characterization of spherical Fe_3_O_4_@Au MNPs (9 ± 1 nm) were described previously in our previous study [41]. Briefly, a two-step process was carried out to construct a core-shell structured Au–Fe_3_O_4_ hybrid particle, which exhibited the optical properties of Au metal and the magnetic properties of Fe_3_O_4_. Firstly, pre-prepared Fe_3_O_4_ nanoparticles were homogeneously dispersed in an EDTA solution. Afterwards, in the presence of the mixture of surfactant solution CTAB and gold chloride salt, the hydroxylamine solution was used to reduce Au^3+^ into Au° and gold was reactively deposited onto the EDTA-immobilized Fe_3_O_4_ particles. In order to stabilize and obtain a smoother and homogenous gold layer, gold precursor ions were directly reduced by hydroxylamine. Fe_3_O_4_ and Fe_3_O_4_@Au nanoparticles were fully characterized with UV–Vis spectrum and TEM images [41]. 

### 2.6. Surface Modification Processes of Au NRs and MNPs

Initially, Au NRs were modified with 4-ATP molecules. For this purpose, the required amounts of solid 4-ATP molecules were added to ethanolic Au NR solution to obtain the 20 mM final concentration and kept overnight to form covalent bonds between the Au surface and thiol groups. After washing with MES buffer three times to remove excess 4-ATP molecules, avidin solution was added in the presence of EDC/NHS and the solution was shaken for 30 min before the centrifugation of the solution was performed to remove excess EDC/NHS molecules. Biotin-conjugated rabbit anti-*E. coli* polyclonal antibodies were added and shaken for 30 min after washing the solution with MES buffer three times. The final solution was redispersed in PBS buffer (pH 7.4) after the washing procedure was repeated at least twice.

In the modification process of the MNPs, 11-MUA, which was prepared in the ethanol, was added to Fe_3_O_4_@Au MNPs and kept at room temperature overnight to modify Fe_3_O_4_@Au MNPs with 11-MUA. The excess 11-MUA molecules were discarded from the working solution by applying the washing procedure at least three times with the help of a magnet. In order to activate carboxylic acid groups of 11-MUA, EDC/NHS was added into the solution. Subsequently, avidin solution was added after the washing procedure was applied at least twice. Ethanolamine solution was added to avoid a non-specific interaction. Then, biotin-conjugated anti-*E. coli* antibodies were added into the MNP solution and shaken for 30 min after the washing procedure. The washing procedure was repeated again at least twice, before the final solution was redispersed in PBS buffer (pH 7.4). This biotinylated antibody recognizes all ‘O’ and ‘K’ antigenic serotypes of *E. coli*.

### 2.7. Preparation of Samples for TEM Measurements

A few microliters of MNP/NR solution was dropped onto the formvar-carbon-coated TEM grids and dried at room temperature. Then, measurements were taken by JEOL JEM 1400 TEM at 80 kV. For the preparation of MNP/NR/*E. coli* conjugates, the analyte was dropped on formvar-carbon-coated copper TEM grids, negatively stained with 2% uranyl acetate, and dried at room temperature. Finally, the sandwich immunoassay structures were examined by TEM.

### 2.8. Capillary-Driven Microfluidic Chip

The microfluidic chip shown in Figure 1 was designed to comprise four successive chambers with their own inlet and outlet ports, so that different solutions can be loaded independently to each chamber using a pipette, in the order starting from the leftmost one in Figure 1. Then, 1 mm-deep chambers were connected to each other via 400 µm-wide and 200 µm-deep microchannels. The acute opening and the intersection of the microchannels and the chambers generate a capillary pinning barrier, so that the liquid loaded in a previous chamber does not overflow into the successive chamber. The details of the working principles of the microfluidic chip have been described in our previous works [42,43].

The chambers, inlet and outlet ports, and connecting microchannels were fabricated with polymethylmethacrylate (PMMA) by machining using a benchtop CNC milling machine (Proxxon MF70 CNC-Ready, PROXXON GmbH, Föhren, Germany). The features were closed by sealing them with a second unmachined PMMA substrate by thermo-compressive bonding. The bonding process was carried out at 70 °C under 0.5 MPa of compression by using a hot press (MSE LP M4SH10 Hot Press, MSE Teknoloji Co. Ltd., Kocaeli, Turkey). After 10 min., bonding was achieved, the plates were cooled down and the load was removed to release the chip. To facilitate bonding at relatively low temperatures, the substrates were treated under chloroform vapor for 3 min. prior to bonding. 

### 2.9. General Procedure for the Developed Biosensor System

Modified MNPs were added to *E. coli* solutions, which had different concentrations (1 × 10^1^–1 × 10^7^ cfu mL^−1^) in the PBS solution. The bacteria were conjugated with MNPs and then they were separated using a magnet. In order to remove unbound bacteria, the washing procedure was applied three times with PBS buffer. The MNP-*E. coli* conjugates were placed into the first chamber of the microfluidic chip. The second chamber was filled with *E. coli* antibody-modified Au NRs. The third chamber was filled with PBS solution to apply the washing procedure to the chip. The SERS measurements were taken at the last chamber of the chip. The movement of the formed conjugates from one chamber to the other was provided by using a magnet.

For the quantification studies, different concentrations of *E. coli* solutions interacted with modified MNPs. Then, the same above-mentioned measurement procedure was applied to obtain SERS measurements.

### 2.10. SERS Measurements for E. coli Detection

Different amounts of Au NRs and MNPs were studied to obtain more sensitive results. The experimental conditions were optimized for the Au NR amount for Raman labeling and MNP amount for IMS. The SERS signal intensity increased with the amounts of Au NRs and MNPs, as expected. However, after a certain amount, the microchannels between chambers became stacked, due to the aggregation of the formed sandwich immunoassay structures. This prevents the transportation of the formed sandwich complexes to the measurement chamber. The fabricated chips were initially tested to verify the operation of the capillary valves. During the magnetic particle tests, it was observed that there were traces of magnetic particles stuck on the channel and chamber surfaces. In order to solve this problem, hydrophilicity of the chips was improved by exposing them to air plasma for 1 min after the bonding process. After each use, the microchips were cleaned in water by sonication for 10 min. Then, ethanol was passed through the microchambers with a pipette and dried at 40 °C for 12 h. A microchip can be used more than ten times.

The SERS measurements of each concentration of *E. coli* were recorded. The calibration curve was constructed by using the average of five parallel readings of SERS signal intensity versus logarithmic *E. coli* concentrations. To prove the linearity of this curve, the coefficient of determination was calculated. The theoretical limit of detection was calculated as 7 cfu/mL, using the equation S_LOD_ = S_bl_ + 3 × s_bl_, where S_bl_ is the mean of 10 cfu/mL *E. coli* measurements and s_bl_ is the standard deviation of 10 cfu/mL *E. coli* measurements. The minimum detectable signal (S_LOD_), calculated from the equation, was converted to population density by using a calibration curve constructed with a series of standards. The specificity of the developed method was tested by using different bacteria instead of our analyte, namely *S. enteritidis* and *S. aureus*. The SERS signal intensities for each bacterium were compared with the blank measurement and *E. coli* measurements that have the same concentration.

Experiments were also carried out with skim milk to demonstrate the applicability of the developed sensor system for the detection of *E. coli* in real samples. Prior to analysis, the absence of *E. coli* in the skim milk samples was confirmed by cultural methods. Then, different concentrations of *E. coli* (1.2 × 10^2^ and 1.2 × 10^3^ cfu/mL) were added to skim milk. These spiked samples were analyzed with the developed chip system and the numbers of *E. coli* in the samples were determined. The recovery rate was calculated for each concentration as the ratio of the amount detected by the biosensor system to the concentration that initially spiked. Three repetitions were performed for each experiment. The average of all readings (in total 15 for each concentration) was used.

## 3. Results and Discussions

The sandwich assay was employed to enhance the lowest detection limit of SERS by improving detection sensitivity. This type of sensor was successfully utilized to detect complex matrices. In the sandwich assay, each analyte molecule “sandwich type” between the primary bio-recognition element became immobilized on the sensor’s surface and then was injected into the secondary biorecognition element. The biotin–avidin interaction has been widely employed as a marker of molecular recognition for the immobilization of biomolecules on the sensor’s surface. Furthermore, by modifying avidin or streptavidin, biotin-modified antibodies can be easily immobilized, and capture target analytes detected in the avidin-biotin detection system.

MNPs also serve a critical role in the biosensor’s function and quality, and owing to the presence of the magnetic field, the sample concentration is increased as well as the sample purity [44]. Fe_3_O_4_ nanoparticles engage in target surface immobilization because of their oxidative stability, compatibility in aqueous environments, and their nontoxicity and suitable groups at the superficial surface. Additionally, unlike conventional purification procedures, the immobilized biomolecules on MNPs may be collected or dispersed in solution using an external magnetic field, allowing for fast sample preparation. In this study, *E. coli*-specific antibodies were immobilized on the MNPs’ surfaces to capture *E. coli* from the sample matrix. Then, *E. coli*-specific antibody-modified Au NRs interacted with the MNP-*E. coli* structures for SERS labeling. The avidin–biotin interaction was used to bind antibodies to the nanoparticle surface. After *E. coli* was captured by the modified MNPs with 30 min of shaking time, the modified Au NRs were attached to the *E. coli* surface in the microfluidic chip. SERS measurements were recorded in the last chamber after the washing step in the chip. It is important to mention that usage of a microfluidic chip, except for the isolation of *E. coli* from solution, allows us to benefit from the advantages of miniaturized systems compared to the conventional batch technique.

### 3.1. General Properties of the Chip and General Procedure for Sandwich Complex Formation

The main aim of the study was to develop a simple sensor device for the detection of pathogens in the milk matrix by combining the capillary driven microfluidic chip and the SERS measurements using MNPs conjugated with antibodies. For this purpose, in the present study, the avidin-biotin modification of the MNPs and Au NRs’ surface was performed. After the surface modifications, the target pathogen *E. coli* was selectively captured by using antibody-modified MNPs in the first chamber. Then, the target *E. coli* was labelled with SERS probe Au NRs in the second chamber of the capillary-driven microfluidic chip. Subsequently, the pathogen–nanoparticle complex is moved to the third chamber by applying a simple magnet for washing with buffer solution in the chip and the target pathogen detection was performed using SERS measurements at the last chamber of the capillary-driven microfluidic chip. In this way, the analytical validation parameters of the method were determined using the proposed sensor platform. The operation of the chip was solely based on the meniscus pinning effect at the intersection of the channels that connected the chambers and the successive chambers. The meniscus pinning phenomenon requires the wettability, which is characterized by the contact angle between the working liquid and the structural solid material, of the material to be stable and predictable during the operation. However, a very well-known disadvantage of PDMS is that it does not have stable surface wettability [45,46]. Therefore, instead of PDMS, we preferred to use PMMA, whose wettability is predictable, with a contact angle of water on PMMA of about 70 degrees, and is controllable [47].

Besides the wettability issues, PMMA also simplifies the fabrication process, as it might be possible to manufacture the chip in a single run of a CNC milling machine. If PDMS was selected instead, the fabrication would require manufacturing of a mold, which would include stepped features as the depth of the chambers and channels are different on our chip. In typical PDMS molding processes, this could be obtained by two successive lithography operations, which would be costly in comparison to milling.

The developed capillary-driven microfluidic chip can be an alternative method to the expensive batch type techniques that necessitate the consumption of excess amounts of sample and materials. Figure 1 displays the overall strategy of the SERS-based detection of *E. coli*.

### 3.2. Characterizations of Au NRs and MNPs

The UV–Vis spectrum of Au NRs is shown in Figure 2A, which indicates absorption peaks at 520 nm and 740 nm. The aspect ratio of gold nanoparticles is measured manually based on the TEM images. The TEM image of the Au NRs indicates that the length of the major axis of gold nanorod particles is 45 ± 3 nm and the length of the minor axis is 15 ± 3 nm (corresponding aspect ratio of 3), as shown in Figure 2B, and homogeneous distribution of the Au NRs was also observed. Detailed characterization of Fe_3_O_4_@Au MNPs with TEM images, X-ray diffraction, magnetic field dependence of MNPs, AFM and UV–Vis spectroscopy results was presented in our previous study [41].

In order to confirm the binding of *E. coli* to MNPs and Au NRs in the chip system, TEM measurements were conducted. After the washing steps, MNP–antibody–*E. coli* conjugates were pipetted into the first chamber of the chip. Sandwich complexes were transformed to the second chamber, in order to label *E. coli* with Au NRs, which were specific to *E. coli*, by the interaction between the antibody-modified Au NRs and *E. coli* surface. Figure 3 indicates the TEM images of *E. coli* interactions with antibody-modified MNPs and Au NRs. The TEM images demonstrated the immobilization of the MNPs on bacteria at high amplifications and both nanoparticles covered some part of the target *E. coli* membrane. Magnetic gold nanoparticles, synthesized by using the current system, showed strong magnetism, and after interaction between MNP and *E. coli*, the effective separation of the MNP–antibody–*E. coli* conjugates was achieved using only a magnet. We demonstrated that IMS percentages varied in a range of 52.1–21.9% and were dependent on the initial bacteria counts [41]. Generally, antibody-modified particles have been used in IMS and resulted in low immobilization efficiency. However, in the present study, we applied oriented antibody–antigen interactions and the obtained results demonstrated high immobilization efficiency after specific interaction.

### 3.3. Optimization of Analytical Parameters and Real Sample Application

When the amounts of Au NRs and MNPs were increased to a certain amount, the SERS signal intensities also increased, as expected. However, the microchannels were clogged after a certain amount of nanoparticles due to the formation of larger aggregates. This prevents the movement of formed sandwich complexes, with the help of a magnet, to the measurement chamber. SERS measurements were performed by interactions of optimum amounts of modified Au NRs and MNPs with different concentrations of *E. coli* solutions. Since 4-ATP has a strong affinity to gold nanoparticles, this molecule can be adsorbed on the surface via Au-S bonds. As revealed from the SERS spectra, two strong peaks at 1080 and 1590 cm^−1^ can be observed, while the peaks at 1080 and 1590 cm^−1^ can be assigned to the a1 modes of 4-ATP molecules on the metal surface [48]. Figure 4A,B show the increasing SERS signal intensities with the increasing amounts of modified MNPs and Au NRs. Each data point is the average of three parallel measurements. Optimum amounts were determined as 2.0 × 10^12^ particles/mL for AuNRs and 5.0 × 10^12^ particles/mL for MNPs, respectively.

### 3.4. Analytical Performance of the Method

Under the optimized conditions, SERS intensity increases with *E. coli* population density. SERS measurements were recorded by interacting optimum amounts of antibody-modified MNPs and Au NRs with differently concentrated *E. coli* solutions, ranging from 10^1^ to 10^7^ cfu/mL, and the obtained spectra are shown in Figure 5. It is important to mention that the blank measurement has a small signal intensity, and the SERS response can be obtained for *E. coli* by subtracting the blank signal intensity from the other signals. Figure 6 shows the calibration curve constructed by using the SERS intensity. A linear relationship between *E. coli* population density and SERS intensity was observed in the range of 10^1^–10^7^ cfu/mL, with a high R^2^ value of 0.9929. The correlation equation was y = 480x − 437, where y is the SERS intensity and x is the population density of *E. coli*. As shown in Figure 5 and Figure 6, the SERS signal intensity increases with the increase in *E. coli* population densities. The observed sensitivity of the developed method for the detection of bacteria can be attributed to the use of stagnant fluid that results in high integration time in the microfluidic chip system.

Furthermore, the method was tested by utilizing the same procedure for different types of bacteria to be sure about the specificity of the developed method to *E. coli*. Here, we decided to utilize a middle concentration value and 10^4^ cfu/mL population density of each bacteria species should be selected to perform experiments. The selectivity of the method for *E. coli* was evaluated with 1 × 10^4^ cfu/mL of *S. aureus* and *S. enteritidis*. Firstly, MNP conjugated to the *E. coli* selective antibody interacted with those bacteria in a separate vial. Secondly, the same experiment was repeated in the presence of 1 × 10^4^ cfu/mL *E. coli*. SERS signals obtained from 1 × 10^4^ cfu/mL of *S. enteritidis* and *S. aureus* were compared with blank signals and spiked samples were compared with the 1 × 10^4^ cfu/mL *E. coli* standard signal. Three parallel microchips were used for each experiment. As shown in Figure 7, the SERS signal remained unchanged, which was the same for the blank signal, for *S. enteritidis* and *S. aureus* and the SERS signal intensities of *S. enteritidis* and *S. aureus* solutions (1 × 10^4^ cfu/mL), both fortified with 1 × 10^4^ cfu/mL *E. coli,* were the same as the standard that contained only 1 × 10^4^ cfu/mL *E. coli.* The obtained SERS signal intensities for *S. enteritidis* and *S. aureus* were quite similar to the blank signal intensity value, which is shown in Figure 7. These SERS results demonstrated the high selectivity of the proposed method in the capillary-driven microfluidic chip for the *E. coli* assay.

The applicability of the developed method to real samples was also proved by the recovery experiments with *E. coli* spiked milk samples, since no detectable *E. coli* was found in the original samples. Two different concentrations of *E. coli* were added to the milk samples and analyzed with the developed system. Since the formed sandwich complexes were washed in PBS buffer before the measurement chamber, the effect of milk on the SERS spectra is ignored, due to the high intensity of the 4-ATP spectrum. The results calculated by using the calibration curve equation were compared with the known initial bacteria counts. A good agreement with 113–120% recovery values are shown in Table 1. In other words, the quantification of *E. coli* in milk samples can be reliably obtained by using the developed method.

In our previous studies, we generated quantum dot nanoparticles (QDs) as labels and fluorescence measurements were described in order to evaluate the applicability of the method using tap water and lake water samples [17,19]. The main limitation in the fluorescence-based immunoassay is the time-consuming modification steps of QDs. In the present study, we applied SERS measurements and conjugation of Au NRs to the MNP- *E. coli* and the washing step was performed in the passive chip and the time necessary for washing away the excess gold nanorods decreased to 1–2 min. Furthermore, the detection limit was low using the SERS measurements compared to the bacteria detection methods and sandwich-type studies based on the use of microchips in the literature. The capillary-driven microfluidic chip can be used as a disposable device and presents excellent robustness. It has, thus, high potential in the rapid analysis of bacterial samples, particularly if combined with portable Raman instruments. Table 2 shows a comparison between the developed immunoassay system and other studies in the literature in terms of total analysis time, the LOD values and dynamic ranges.

## 4. Conclusions

The rapid detection of pathogenic bacteria has important significance for food safety production, which enables manufacturers to make rapid adjustments according to the detection results. While conventional microbial culture testing has long been the most common method for enumerating bacteria, it has some limitations, such as multi-day incubation periods and inability to detect viable but not culturable bacteria. Although MNPs are effective at detecting bacteria, researchers find it difficult to detect low quantities of target bacteria in real samples; therefore, MNPs must include additional materials for surface modification. In addition to the excellent LOD value and short analysis time advantages of the proposed immunoassay system, selective analysis is also possible due to usage of *E. coli*-specific antibodies. Furthermore, consumption of sample volumes, chemicals and nanoparticles is minimized due to the usage of microfluidic measurement platforms.

## Figures and Tables

**Figure 1 biosensors-12-00765-f001:**
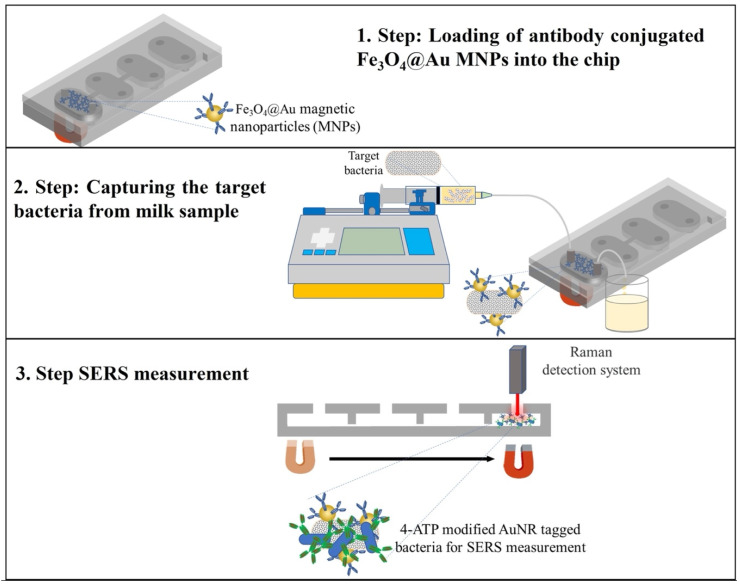
The schematic illustration of experimental strategy using SERS-based sandwich immunoassay for detection of *E. coli*.

**Figure 2 biosensors-12-00765-f002:**
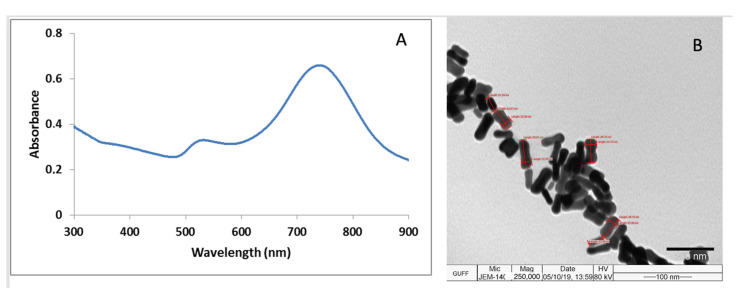
(**A**) UV–Vis spectrum of Au NRs. (**B**) TEM images of Au NRs.

**Figure 3 biosensors-12-00765-f003:**
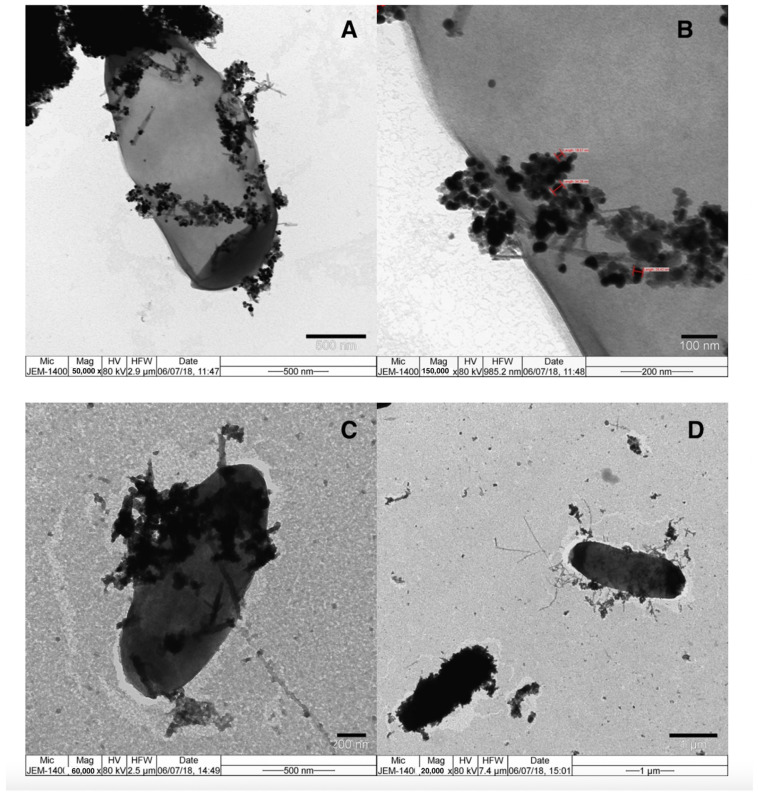
TEM images of *E. coli* interactions with modified MNPs (**A**,**B**) and MNPs with Au NRs (**C**,**D**).

**Figure 4 biosensors-12-00765-f004:**
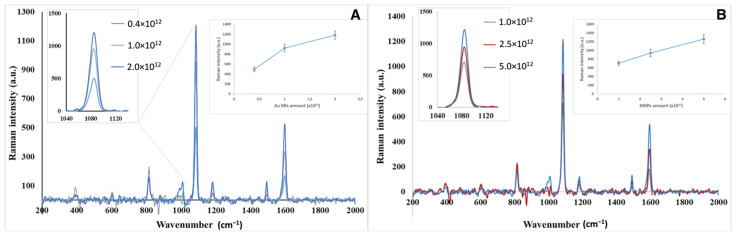
(**A**) SERS spectra obtained for Au NR amount optimization; (**B**) SERS spectra obtained for MNP amount optimization.

**Figure 5 biosensors-12-00765-f005:**
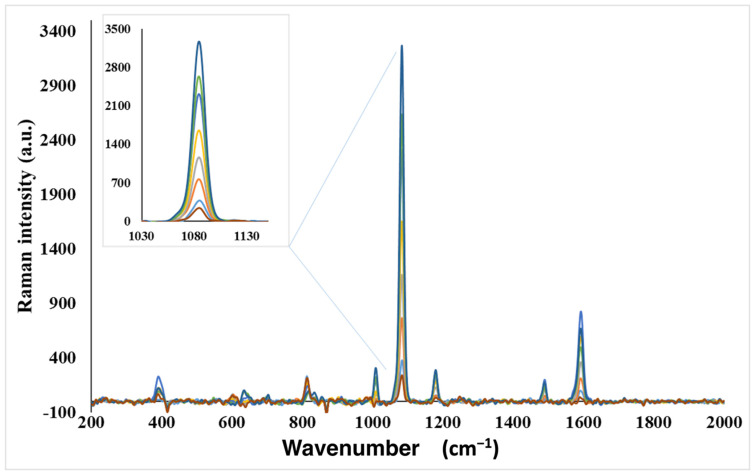
Typical SERS spectra obtained from target bacteria as a function of different initial logarithmic concentrations. Different color represents the increase of bacteria concentration.

**Figure 6 biosensors-12-00765-f006:**
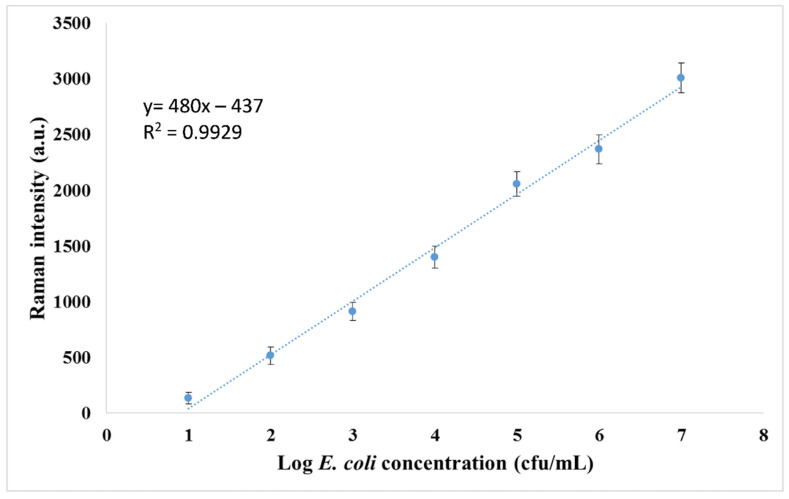
Calibration curve obtained for *E. coli*.

**Figure 7 biosensors-12-00765-f007:**
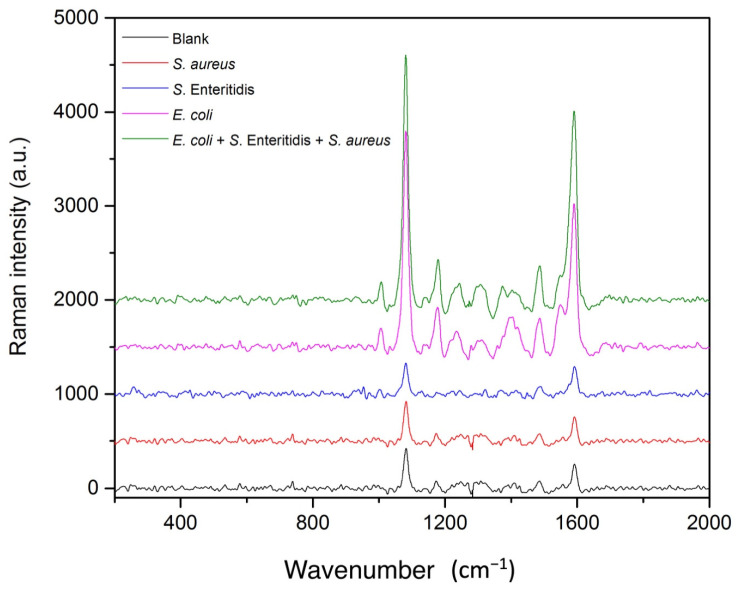
SERS signal intensities measured for blank, *S. aureus*, *S. enteritidis*, *E. coli*, and *S. aureus* and *S. enteritidis* spiked with 1 × 10^4^ cfu/mL *E. coli*, respectively.

**Table 1 biosensors-12-00765-t001:** Comparison of the results obtained from the spiked milk samples.

Spiked *E. coli* Concentration (cfu/mL)	Detected *E. coli* Concentration (cfu/mL)	Recovery (%)
1.2 × 10^2^	1.4 ± 0.3 × 10^2^	113
1.2 × 10^4^	1.5 ± 0.4 × 10^4^	120

**Table 2 biosensors-12-00765-t002:** Comparison of this proposed biosensor with other similar studies in the literature.

Total Analysis Time	LOD (cfu mL^−1^)	Dynamic Range (cfu mL^−1^)	References
30 min	2.9 × 10^2^	10^1^–10^5^	[49]
20 min	10^1^	10^2^–10^6^	[50]
2 h	1	10^0^–10^6^	[51]
Less than 1 h	5 × 10^2^	10^3^–5 × 10^7^	[52]
5 min	1.2 × 10^1^	10^1^–10^5^	[53]
20 min	2.5 × 10^2^	5 × 10^2^–5 × 10^3^	[54]
Less than 70 min	8	10^1^–10^4^	[55]
3 h	10^5^	10^5^–10^7^	[56]
Less than 1 h	7	10^1^–10^7^	This study

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
