# Peer review of "Escherichia coli Enumeration in a Capillary-Driven Microfluidic Chip with SERS"

_biosensors, 2022, doi:10.3390/bios12090765_

Round 1

Reviewer 1 Report

This paper presented a developed microfluidic approach with the integration of immunomagnetic separation and SERS to sensitively detect E.coli with specificity to some extent. When compared with the article published in the Analytical method by the same group, this paper adopted SERS instead of fluorescence, which was claimed to offer time-effectiveness and high sensitivity. Overall I recommend that this draft is suitable to be published on Biosensors with some required or minor improvements.

1. Authors should put the current Figure 2 as Figure 1, to show the experiment schematics first. And probably make the Au NR characterization as supplementary information. Especially you already mentioned Figure 2 first in the text.

2. Line 61, "reduced reliability" sounds too general. Can authors give some specific comparison, in regard to cost, labor, time, etc?

3. Line 199-200. Have authors tried to avoid this aggregation, for example, by coating, changing channel geometries or material, external flushing, gas valve, vacuum, pump, etc? At least please add some discussion on the potential solutions.

4. Can the authors validate the reasons why PMMA is superior to PDMS in this application?

5. Figure 4 did not show how the Au NRs and MNP concentrations were optimized to gain the optimal value. Can the authors show the trend containing various tested values like Figure 6? Can authors give an analysis of which factors influence the most? Or assumptions.

6. To test the specificity for Ecoli, can you input mixed samples and prove that the signal for e.coli is still pronounced. In addition, can the authors provide a P-value analysis for Figure 7? Is the SERS signal for S.aures significantly higher than that for blank and S.enterica.

Other formatting and grammatical improvements:

1.Please pay attention to the grammar and symbols used in the paper.

e.g., line 50, the removal of 88% Escherichia coli (E. coli) from the sample have (been) achieved….antibodies have (been) used to….. This kind of erros can be found quite a few.

2.Commas were missing in some places. 

3.Please also take care of the formatting of figures, making fonts, font size, line width, etc uniform across the paper.

Author Response

Referee 1:

Comment 1:  Authors should put the current Figure 2 as Figure 1, to show the experiment schematics first. And probably make the Au NR characterization as supplementary information. Especially you already mentioned Figure 2 first in the text.

Answer 1: The Figure 2 was changed as Figure 1 in the revised manuscript and the Au NR characterization has mentioned first in the text. 

Comment 2:  Line 61, "reduced reliability" sounds too general. Can authors give some specific comparison, in regard to cost, labor, time, etc?

Answer 2: The following statement and the related reference was also added to the introduction section to explain better this issue.

“Although a few continuous flow type-based microfluidic devices have already demon-strated high sensitivities in the determination of biological relevant organisms, these approaches require long analysis time and do not include in-field analysis strategies of target pathogens [35,36]. A circular dielectrophoretic microfluidic device was also utilized for pathogen analysis from human blood in less than 1 min [37]. Krafft et. al. low-cost disposable PDMS device for the concentration of bacteria from drinking tap water, which function combines filtration with the electrodriven flow [38]. However, these techniques require sophisticated technologies or external apparatuses hardly compatible with a portable device.”

Comment 3: Line 199-200. Have authors tried to avoid this aggregation, for example, by coating, changing channel geometries or material, external flushing, gas valve, vacuum, pump, etc? At least please add some discussion on the potential solutions.

Answer 3: The fabricated chips were initially tested to verify the operation of the capillary valves. During the magnetic particle tests, it was observed that the traces of magnetic particles stuck on the channel and chamber surfaces. In order to solve this problem, hydrophilicity of the chips was improved by exposing them to air plasma for 1 min after the bonding process. After each use, the microchips were cleaned in water by sonication for 10 min. Then, ethanol was passed through the microchambers with a pipette and dried at 40 °C for 12 h. A microchip could be used more than ten times.

The latter statement has been now added to 2.10 in the revised manuscript.

Comment 4:  Can the authors validate the reasons why PMMA is superior to PDMS in this application?

Answer 4 : To provide more insight into the device construction with PMMA, the following statement was added to section 3.1 in the revised the manuscript.

“The operation of the chip is solely based on the meniscus pinning effect at the intersection of the channels connecting the chambers and the successive chambers. The meniscus pinning phenomenon requires the wettability, which is characterized by the contact angle between the working liquid and the structural solid material, of the material be stable and predictable during the operation. However, a very well-known disadvantage of PDMS is that it does not have a stable surface wettability [45,46]. Therefore, instead of PDMS, we preferred to use PMMA, whose wettability is predictable with a contact angle of water on PMMA as about 70 degrees and controllable [47].

Besides the wettability issues, PMMA also simplifies the fabrication process, as it could be possible to manufacture the chip in single run of a CNC milling machine. If PDMS were selected instead, the fabrication would require manufacturing of a mold, which would include stepped features as the depth of the chambers and channels are different on our chip. In typical PDMS molding process, this could be obtained by two successive lithography operations, which would be costly in comparison to milling.”

Comment 5:  Figure 4 did not show how the Au NRs and MNP concentrations were optimized to gain the optimal value. Can the authors show the trend containing various tested values like Figure 6? Can authors give an analysis of which factors influence the most? Or assumptions.

Answer 5: The trend containing various tested values like Figure 6 has added for Figure 4. An assumption for the analysis of which factors influence the most the sensitivity is added in the text.

Comment 6:  To test the specificity for E.coli, can you input mixed samples and prove that the signal for E.coli is still pronounced. In addition, can the authors provide a P-value analysis for Figure 7? Is the SERS signal for S.aures significantly higher than that for blank and S.enterica.

Answer 6: In the present study, the E. coli signal was recorded using standard addition method in the presence of S. Enteritidis and S. aureus. The selectivity of the method for E. coli was evaluated with 1×104 cfu/mL of S.aures and S. Enteritidis. Firstly, MNP conjugated to the E. coli selective antibody was interacted with those bacteria in separate vial.  Secondly, same experiment was repeated in the presence of 1×104 cfu/mL E. coli.  SERS signals obtained from 1×104 cfu/mL of S.aures and S. Enteritidis were compared with blank signal and spiked samples were compared with 1×104 cfu/mL E.coli standard signal. Three parallel microchip was used for each experiment. As shown in Fig. 7, the SERS signal remained unchanged, same as blank signal, for  S.aures and S. enteritidis These results suggest the high selectivity of the proposed method for E. coli assay.

Other formatting and grammatical improvements:

1.Please pay attention to the grammar and symbols used in the paper.

e.g., line 50, the removal of 88% Escherichia coli (E. coli) from the sample have (been) achieved….antibodies have (been) used to….. This kind of erros can be found quite a few.

We checked the entire manuscript for grammatical errors as the Referee requested.

2.Commas were missing in some places. 

It is checked, corrected in the text. 

3.Please also take care of, making fonts, font size, line width, etc uniform across the paper.

The entire manuscript is checked, corrected in the revised manuscript. 

Reviewer 2 Report

The manuscript of Uzeyir Dogan et al. entitled ‘Escherichia coli enumeration in a capillary driven microfluidic chip with SERS’ focuses on development a simple sensor device for the detection of E.coli  in the food matrix by combining the capillary driven microfluidic chip and SERS measurements using MNPs conjugated with antibodies. It is an interesting approach. The manuscript and its division into sections and subsections are well organized. Apart from that, the manuscript is also very well written and should be understandable to all. The manuscript is in the scope of Biosensors Journal, but before acceptance of this manuscript for publication some additional measurements, corrections are necessary and some explanations should be provided. The main problem in my opinion is the representativeness of the results obtained and the presentation of the data. The following are my detailed comments on the manuscript.

Major comments:

1) The issue of getting results quickly for the food and cosmetics industry is debatable in my opinion. Most ISO standards are based on the use of biochemical methods, including the cultivation of bacteria on selective solid media, e.g. chromagar, dedicated to specific bacterial species. They are cultured for 24 to 48 hours. Thus, the main requirements of the food or cosmetics industry are, for example, to facilitate or streamline sample preparation procedures or to eliminate the involvement of qualified microbiological personnel. This method does not offer these facilities. The question of detection time is most relevant, however, in the case of medical microbiological diagnostics, where sometimes every hour counts and diagnostic tests are always performed by qualified laboratory personnel. The authors should therefore think more about the real benefits that this technique can offer and where it can be used.

2) The authors claim that their solution can be used effectively in the food industry. Completely? In my opinion, this is also questionable, because in the food industry we don’t have only liquid products such as milk but also the solid products e.g. meat-based. The proposed method can be used also for this kind of samples? I don’t think so according to the proposed purification procedure, which will be insufficient for such samples.

3) As stated by the authors, the main goal of the proposed biosensor is rapid pathogen detection; however, in the introduction section they analyzed merely other alternative techniques according to the recent state of art. In the Introduction section, the authors have indicated some conventional methods for identification of Escherichia coli, such as ELISA, electrochemical, fluorescent, colorimetric and SPR techniques. The authors did not analyze other alternatives including the aptamer-based biosensors, luminescent-phage based biosensors, colonies-scattering-based optical biosensors or even smart phone-based biosensors,  which were also used for the detection and identification of E.coli. In my opinion, these alternatives should indicated and compared to highlight the advantages of the proposed technique (related to the time of detection/identification, complexity of sample preparation, sensitivity/accuracy, costs, etc.) as some of these techniques are based on the examination of samples prepared in a much simpler way, allowing their use by nonscientific/nontechnical personnel (see examples bellow):

https://doi.org/10.1016/j.snb.2017.09.121

https://doi.org/10.1111/1541-4337.12908

https://doi.org/10.1016/j.aca.2019.07.028

https://doi.org/10.1021/acsami.0c00418

https://doi.org/10.1063/5.0057787

https://doi.org/10.1016/j.bios.2020.112761

https://doi.org/10.1016/j.bios.2017.08.002

https://doi.org/10.1016/j.bios.2018.06.005

https://doi.org/10.1016/j.snb.2017.12.110

https://doi.org/10.1016/j.yofte.2018.09.012

4) I wonder why the examination was performed only on a single E.coli strain? The fundamental concept of these kind o sensors is based on antibody-modified MNPs which are responsible for the capture of the bacteria cells. However, it is known that because of the genetic similarity among serovars/strains, antibodies or nucleic acid probes show cross-reactions that limit the identification of E.coli serovars. On what basis then do the authors conclude that the developed sensor will allow the detection of different strains of E. coli? The presented results do not demonstrate or confirm this. They only show that a given (more precisely, one) strain has been classified as E.coli. Why did the authors not performed the validation of the proposed method with more than one strain of E.coli?

5) Why did the authors not decide to prepare the mixture of different bacteria for examination by the proposed method, but analyzed samples of different bacteria separately? The mixture of bacteria is closest to the real microbiological samples? The contamination of the samples by bacteria monoculture is not so common.

6) In the manuscript, the authors did not indicate which strains of E. coli, S. Enteritidis and S. aureus bacteria were used in the examination performed. Provide a detailed description of these reference stains.

7) How many repetitions of bacteria samples were examined for each concentration? This is not indicated in the manuscript? The authors analyzed the data.

8) Why is centrifugation is necessary during the procedure of the sample preparation procedure? This technique of sample purification is not so common.

9)Lines 262-264: “. The capillary-driven microfluidic chip developed in this way can be an alternative method to the expensive traditional techniques that necessitate the consumption of an excess amount of sample and materials. ‘- any comparison? Please explain. The traditional method is e.g. the cultivation of bacterial colonies in solid nutrient media. The proposed method is cheaper than this mentioned technique?

10) Lines 275-276: “TEM images demonstrated the immobilization of the MNPs on bacteria at high amplification and both nanoparticles covered the some part of target E.coli membrane.” – any quantitative measure?

11) Line 203-205: “The calibration curve was constructed by using the average of five parallel readings of SERS signal intensity versus logarithmic E. coli concentrations.”- why so few measurements? This allowed for the elimination of noise and proper averaging at all? What measure of uncertainty was used to determine the mean Raman intensity value for a given concentration of bacteria? The standard deviation for 5 measurements is not very meaningful.

12) Fig.4: not specifying which colour corresponds to which concentration of bacteria. In recent form, these spectra are not readable.

13) It is possible to obtain the negative value of the Raman intensity (as in Fig.4 and 5) or any other intensity at all? Probably, as indicated in lines 296-297: ‘Each data point is the average of three parallel measurements’ this was due to an insufficient number of repeat measurements, which did not allow effective averaging of the results and elimination of noise? Why did the authors not perform more repeated measurements? Did you extract the background/base line? Why did the authors choose to use the maximum Raman intensity as a measure of bacterial concentration rather than the ratio of the maximum intensities of the two bands?

14) Lines 243-244: “TEM image of Au NRs indicates the aspect ratio is about 3 and homogeneous distribution of Au NRs was also observed.”- please explain.

15) Fig.6/Fig.7: how were the measurement uncertainties of the individual points on this graph determined? What measure has been used here? How many measurements were made? How many repetitions of bacteria samples were examined?

16) Lines 304- 306: “SERS measurements were recorded by interacting optimum amounts of antibody  modified MNPs and Au NRs with differently concentrated E. coli solutions ranging from 101-107 cfu/mL and the obtained spectra is shown in Figure 5. ‘- why only this range of concentration was used in this examination, why not 108, 109, etc. or concentration slower than 101 CFU/ml?

17) Lines 324-327:” Furthermore, the method was tested by utilizing the same procedure to different type of bacteria to be sure about the specificity of the developed method to only E. coli. The SERS signal intensities for S. enteritidis and S. aureus were quite similar to the blank signal intensity value shown in Figure 7.”- why 104 CFU/ml of each bacteria species used in this examination?

18) Fig. 2: The following stages of the test, particularly their sequence, are not very well visualized in the diagram in Fig.2

19) Table 1: Validation of the proposed method based only on two measurements is not sufficient, in my opinion. Indicate the meaning of ss? How many repetitions were performed in this case?

20) Line 338: “. A good agreement with the recovery values are shown in Table 1”. To indicate the of the proposed method sensitivity the more appropriate Cohen’s h coefficients or more accurately the Kappa statistic can be determined, which are commonly used in diagnostics.

21) In conclusion, it was stated that ‘the results of conventional microbial culture can be obtained after 2-3 days’, which is debatable in my opinion. Please provide an explanation according to which techniques it is related. Moreover, in the introduction it was stated that ‘Classical incubation (24-48 h)” it is related to the sentence cited from conclusions or it is related only to the step of sample preparation step?

22) Lines 356-359:“Proposed approach can be a pioneering work for the isolation and pre-enrichment of pathogens from the food matrix and quantitative detection of pathogens using SERS measurements in a capillary-driven microfluidic chip”- the authors preformed experiments only on contaminated milk, liquid samples. There are no results for solid food samples; therefore, such a general statement is not justified in my opinion.

Minor comments:

1) In the abstract, E.coli should be in italics.

2) Line 38, what do you mean by classical incubation? Please explain

3) Line 45: remove a single citation from this line/raw.

4) Figures should be placed closest to the first mention in text. Why is Fig.2 is mentioned on page 4, before Fig. 1, and the Fig. 2 is placed on page 7? Correct this.

5) Line 242: ‘UV-vis’?

6) Fig. 1a: Add ‘absorbance [au]’ Indicate the central wavelengths indicated in line 243 as related to the maximum of absorbance bands.

7) Fig.1b: with which distance the scale bar is related to? It is not clearly indicated in this figure. The same is true in Fig.3

8) It should be not S. enteritidis, but rather S. Enteritidis as I remember.

Author Response

Referee 2:

Comment 1: The issue of getting results quickly for the food and cosmetics industry is debatable in my opinion. Most ISO standards are based on the use of biochemical methods, including the cultivation of bacteria on selective solid media, e.g. chromagar, dedicated to specific bacterial species. They are cultured for 24 to 48 hours. Thus, the main requirements of the food or cosmetics industry are, for example, to facilitate or streamline sample preparation procedures or to eliminate the involvement of qualified microbiological personnel. This method does not offer these facilities. The question of detection time is most relevant, however, in the case of medical microbiological diagnostics, where sometimes every hour counts and diagnostic tests are always performed by qualified laboratory personnel. The authors should therefore think more about the real benefits that this technique can offer and where it can be used.

Answer 1: This work presents a model of low-cost and simple SERS-based device for fast analysis of bacteria from milk matrix. Herein, we focused our efforts on the SERS measurement and the investigation of its working principle using bacterial pathogens. We are convinced, that it has a great potential, which will be turned in the future into the desired set-up. We are fully aware that more experiments must be conducted to fully prove this (e.g., the extension of sample volume, investigation of the robustness of the device for various food samples, determination of common validation parameters). However, the aim of this work lied particularly in the demonstration of the remarkable potential of used approach and viability of the constructed device. Furthermore, there is a wide range of portable Raman spectrometers in the market as wells as microfluidic designs for capillary microfluidics.

We added remarks throughout the revised manuscript to point out the goal of our research more clearly (page 11), “The capillary driven microfluidic chip can be used as a disposable and presents excellent  robustness. It has thus a high potential for rapid analysis of bacterial samples, particularly if combined with portable Raman instruments.”

Comment 2: The authors claim that their solution can be used effectively in the food industry. Completely? In my opinion, this is also questionable, because in the food industry we don’t have only liquid products such as milk but also the solid products e.g. meat-based. The proposed method can be used also for this kind of samples? I don’t think so according to the proposed purification procedure, which will be insufficient for such samples.

Answer 2: The referee is right.  The proposed purification procedure can be insufficient for the solid products e.g. meat-based. In order to solve the problem as the referee has mentioned, preliminary steps to extract target bacteria from the sample matrix should be considered.

Comment 3: As stated by the authors, the main goal of the proposed biosensor is rapid pathogen detection; however, in the introduction section they analyzed merely other alternative techniques according to the recent state of art. In the Introduction section, the authors have indicated some conventional methods for identification of Escherichia coli, such as ELISA, electrochemical, fluorescent, colorimetric and SPR techniques. The authors did not analyze other alternatives including the aptamer-based biosensors, luminescent-phage based biosensors, colonies-scattering-based optical biosensors or even smart phone-based biosensors,  which were also used for the detection and identification of E.coli. In my opinion, these alternatives should indicated and compared to highlight the advantages of the proposed technique (related to the time of detection/identification, complexity of sample preparation, sensitivity/accuracy, costs, etc.) as some of these techniques are based on the examination of samples prepared in a much simpler way, allowing their use by nonscientific/nontechnical personnel (see examples bellow): 

https://doi.org/10.1016/j.snb.2017.09.121, https://doi.org/10.1111/1541-4337.12908

https://doi.org/10.1016/j.aca.2019.07.028, https://doi.org/10.1021/acsami.0c00418

https://doi.org/10.1063/5.0057787, https://doi.org/10.1016/j.bios.2020.112761

https://doi.org/10.1016/j.bios.2017.08.002, https://doi.org/10.1016/j.bios.2018.06.005

https://doi.org/10.1016/j.snb.2017.12.110, https://doi.org/10.1016/j.yofte.2018.09.012

Answer 3: A comparison table has been added in the results and discussion part by considering the parameters you have offered and by using the articles you have shared above. 

Table 2 shows a comparison between the developed immunoassay system and with other studies in literature in terms of total analysis time, the LOD values and dynamic ranges.

Table 2. Comparison of this proposed biosensor with other similar studies in the literature.

Total Analysis Time

LOD (cfu mL-1)

Dynamic Range (cfu mL-1)

Reference

30 min

2.9x102

101-105

49

20 min

101

102-106

50

2 hours

1

100-106

51

less than 1h

5x102

103-5x107

52

5 min

1.2x101

101-105

53

20 min

2.5x102

5x102-5x103

54

less than 70 min

8

101-104

55

3 h

105

105-107

56

less than 1h

7

101-107

this study

Comment 4: I wonder why the examination was performed only on a single E.coli strain? The fundamental concept of these kind o sensors is based on antibody-modified MNPs which are responsible for the capture of the bacteria cells. However, it is known that because of the genetic similarity among serovars/strains, antibodies or nucleic acid probes show cross-reactions that limit the identification of E.coli serovars. On what basis then do the authors conclude that the developed sensor will allow the detection of different strains of E. coli? The presented results do not demonstrate or confirm this. They only show that a given (more precisely, one) strain has been classified as E.coli. Why did the authors not performed the validation of the proposed method with more than one strain of E.coli?

Answer 4: The referee is right that the proposed method is specific to E. coli. We indicated serotype of E.coli (ATCC 35218) in section 2.4. The following statement and the related reference was also added to the first paragraph of the section 2.6 to explain better this issue.

“This biotinylated antibody recognizes all ‘O’ and ‘K’ antigenic serotypes of E. coli (https://www.fitzgerald-fii.com/e-coli-antibody-biotin-60-e13b.html (accessed 11 July 2019).”   

Comment 5: Why did the authors not decide to prepare the mixture of different bacteria for examination by the proposed method, but analyzed samples of different bacteria separately? The mixture of bacteria is closest to the real microbiological samples? The contamination of the samples by bacteria monoculture is not so common.

Answer 5: In the present study, in first part of the experiments non-E. coli cultures was used separately at cell densities of 1×104 cfu/mL and the SERS signal was compared with blank signal.  We carried out  experiments with 104 cfu/mL E. coli and SERS signals were compared with 104 cfu/mL E. coli signal.

Comment 6: In the manuscript, the authors did not indicate which strains of E. coli, S. Enteritidis and S. aureus bacteria were used in the examination performed. Provide a detailed description of these reference stains.

Answer 6: In the original we had referred to previous work. In the revision, we provide more detail concerning material and methods. The related statement in section 2.2 was corrected as:

E. coli (ATCC35218), S. enteritidis (ATCC BAA1045) were obtained from Hacettepe University Food Research Center Culture Collection, Ankara, Turkey.”

Comment 7: How many repetitions of bacteria samples were examined for each concentration? This is not indicated in the manuscript? The authors analyzed the data. 

Answer 7: Three repetitions of bacteria samples were examined for each concentration. This is now indicated in the revised manuscript.

Comment 8: Why is centrifugation is necessary during the procedure of the sample preparation procedure? This technique of sample purification is not so common.

Answer 8: Centrifugation is not necessary during the procedure of the milk sample preparation procedure. It was used for to get pure microorganism standards not for the sample preparation. 

Comment 9: Lines 262-264: “The capillary-driven microfluidic chip developed in this way can be an alternative method to the expensive traditional techniques that necessitate the consumption of an excess amount of sample and materials. ‘- any comparison? Please explain. The traditional method is e.g. the cultivation of bacterial colonies in solid nutrient media. The proposed method is cheaper than this mentioned technique?

Answer 9: That statement is true for the batch type techniques which needs more sample volume and usage of more chemicals and nanoparticles. This statement is used for a general comparison of batch type techniques versus microfluidic measurement platform. One of our previous study (ref 17) can be given an example in which the volume of consumed sample, chemicals and nanoparticles were quite higher than this procedure.

The term “traditional” in the related part of the manuscript has been replaced with “batch type techniques”

Comment 10: Lines 275-276: “TEM images demonstrated the immobilization of the MNPs on bacteria at high amplification and both nanoparticles covered the some part of target E.coli membrane.” – any quantitative measure?

Answer 10: The population densities of the standard solution was determined by plate counting method as 1.2x104 cfu/mL. A 500 µL of these solutions were added to 100 mL of sample solutions.  After the treatment with MNP, the E. coli were captured, the supernatant was discarded and diluted to 500 µL with PBS. Therefore, the final population density was 1.2×104 cfu/mL.

Comment 11: Line 203-205: “The calibration curve was constructed by using the average of five parallel readings of SERS signal intensity versus logarithmic E. coli concentrations.”- why so few measurements? This allowed for the elimination of noise and proper averaging at all? What measure of uncertainty was used to determine the mean Raman intensity value for a given concentration of bacteria? The standard deviation for 5 measurements is not very meaningful. 

Answer 11: In section 2.10 sample preparation was described in order to evaluate the applicability of the method to milk samples. The results were presented in Table 1 and discussed in section 3.4.  However, in section 2.9 defines the range for E. coli population density studied in order to determine the linear range for calibration. The population density starting from 1 x 107 cfu/mL and proper dilutions were performed till 1x101 cfu/mL.

In the revision, we provide more detail concerning material and methods. The related statement in section 2.10 was corrected as:

“Three repetitions were performed for each experiment.  The average of all readings (in total 15 for each concentration) was used.”

Comment 12: Fig.4: not specifying which colour corresponds to which concentration of bacteria. In recent form, these spectra are not readable. 

Answer 12: Fig.4 is now redrawn and replaced in the text.

Comment 13: It is possible to obtain the negative value of the Raman intensity (as in Fig.4 and 5) or any other intensity at all? Probably, as indicated in lines 296-297: ‘Each data point is the average of three parallel measurements’ this was due to an insufficient number of repeat measurements, which did not allow effective averaging of the results and elimination of noise? Why did the authors not perform more repeated measurements? Did you extract the background/base line? Why did the authors choose to use the maximum Raman intensity as a measure of bacterial concentration rather than the ratio of the maximum intensities of the two bands?

Answer 13: The background/base line was extracted due to the removal of negative signals. Three repetitions were performed for each experiment.  The average of all readings (in total 9 for each data point) was used. Both approaches can be used to get results as the referee has mentioned.

 The referee is right that the ratio of the maximum intensities of the two bands can be also used. Since our calibration curve was linear and signals shows similar trends for the most intense two bands, we preferred to use the maximum Raman intensity.

Comment 14: Lines 243-244: “TEM image of Au NRs indicates the aspect ratio is about 3 and homogeneous distribution of Au NRs was also observed.”- please explain.

Answer 14: To explain better, a phrase was added to the part 3.2.

“The rod-shaped gold nanoparticles have an average diameter of 15 nm and an average length of about 45 nm (corresponding aspect ratio of 3).”

Comment 15:  Fig.6/Fig.7: how were the measurement uncertainties of the individual points on this graph determined? What measure has been used here? How many measurements were made? How many repetitions of bacteria samples were examined? 

Answer 15: Regarding Fig.6/Fig.7, It was directly calculated by excel program by calculating RSD values of maximum Raman signal intensity of all readings in total 15 for each data point.

Comment 16:  Lines 304- 306: “SERS measurements were recorded by interacting optimum amounts of antibody  modified MNPs and Au NRs with differently concentrated E. coli solutions ranging from 101-107 cfu/mL and the obtained spectra is shown in Figure 5. ‘- why only this range of concentration was used in this examination, why not 108, 109, etc. or concentration slower than 101 CFU/ml?

Answer 16: After 107 cfu/mL bacteria concentration, the calibration curve deviates from the linearity. Furthermore, microchannels between chambers were started to stacked due to aggregation of the formed sandwich immunoassay structures after this bacteria concentration level. This prevents the transportation of the formed sandwich complexes to the measurement chamber which is stated in the text.

Comment 17:  Lines 324-327:” Furthermore, the method was tested by utilizing the same procedure to different type of bacteria to be sure about the specificity of the developed method to only E. coli. The SERS signal intensities for S. enteritidis and S. aureus were quite similar to the blank signal intensity value shown in Figure 7.”- why 104 CFU/ml of each bacteria species used in this examination? 

Answer 17: A statement has now been added to the end of section 3.4 accordingly.

“Here, we decided to utilize a middle concentration value and 104 cfu/ml population density of each bacteria species should be selected to perform experiments.”

Comment 18:  Fig. 2: The following stages of the test, particularly their sequence, are not very well visualized in the diagram in Fig.2.

Answer 18: Fig. 2 was revised according to the Referee’s suggestion

Comment 19:  Table 1: Validation of the proposed method based only on two measurements is not sufficient, in my opinion. Indicate the meaning of ss? How many repetitions were performed in this case?

Answer 19: In the present study the term “ss” means standard deviation and now it is replaced with sd in the text. This values are calculated from the average of 15 data points for each.

Comment 20:  Line 338: “. A good agreement with the recovery values are shown in Table 1”. To indicate the of the proposed method sensitivity the more appropriate Cohen’s h coefficients or more accurately the Kappa statistic can be determined, which are commonly used in diagnostics.

Answer 20: Regarding Table 1, we added the following at section 3.4.

“Since true values are in the confidence intervals at %95 confidence level, a good agreement about the recovery values can be postulated.”

Comment 21:  In conclusion, it was stated that ‘the results of conventional microbial culture can be obtained after 2-3 days’, which is debatable in my opinion. Please provide an explanation according to which techniques it is related. Moreover, in the introduction it was stated that ‘Classical incubation (24-48 h)” it is related to the sentence cited from conclusions or it is related only to the step of sample preparation step?

Answer 21: We thank to Referee for his valuable comment, this part of the paragraph has been revised according to Referee’s suggestion. Following phrase was added in the conclusion part.

“While conventional microbial culture testing has long been the most common method for enumerating bacteria, it has some limitations such as multi-day incubation periods, inability to detect viable but not culturable bacteria.”

The statement in the introduction have been removed.

Comment 22:  Lines 356-359:“Proposed approach can be a pioneering work for the isolation and pre-enrichment of pathogens from the food matrix and quantitative detection of pathogens using SERS measurements in a capillary-driven microfluidic chip”- the authors preformed experiments only on contaminated milk, liquid samples. There are no results for solid food samples; therefore, such a general statement is not justified in my opinion.

Answer 22:

The statement in conclusion part have been removed in the revised manuscript.

Minor comments:

1) In the abstract, E.coli should be in italics. 

It is checked, corrected in the text.

2) Line 38, what do you mean by classical incubation? Please explain

As suggested by the Referee, the term “classical incubation” is replaced as “plate counting” in related parts throughout the text.

3) Line 45: remove a single citation from this line/raw.

We now removed the citation.

4) Figures should be placed closest to the first mention in text. Why is Fig.2 is mentioned on page 4, before Fig. 1, and the Fig. 2 is placed on page 7? Correct this.

This important detail is now corrected.

5) Line 242: ‘UV-vis’?

It is corrected as UV-Vis.

6) Fig. 1a: Add ‘absorbance [au]’ Indicate the central wavelengths indicated in line 243 as related to the maximum of absorbance bands.

Y-axis in Fig. 1 has been rewritten.

7) Fig.1b: with which distance the scale bar is related to? It is not clearly indicated in this figure. The same is true in Fig.3

The scale bars in Fig 1 and Fig. 3 have been now clearly indicated.

8) It should be not S. enteritidis, but rather S. Enteritidis as I remember.

As suggested, we indicated as S. Enteritidis in the revised manuscript. 

Reviewer 3 Report

The authors present a manuscript on the realization of a microfluidic device for the SERS detection of E. Coli bacteria. The detection methodology is based on the interaction between gold nanorods (AuNRs) and magnetic nanoparticles (MNPs) with bacteria cells.

Although the topic is interesting, the manuscript presents several critical aspects concerning the presentation of the results. In particular, the writing is often inadequate, making hard the comprehension of the reported results. Therefore, in the present form, the manuscript is not suitable for publication.

The major general criticisms that must be addressed to reconsider the manuscript after further revision are listed below:

1. A careful proofreading of the manuscript is required.

2. In the Introduction, a brief description of SERS spectroscopy and of its applications in bacteria detection would be appropriate in regard to the broad readership of the journal. See for example: 10.1039/d1nr00708d; 10.1039/d0nr06340a; 10.1007/s00604-021-04885-z.

3. It is well known that gold nanoparticles have also antibacterial properties, see for example: 10.2147/IJN.S134526; 10.1016/j.jcis.2020.07.006; 10.3390/nano11061621; 10.1016/j.jiph.2021.10.007. This aspect should be discussed in the manuscript to highlight the further potentialities of the proposed methodology.

4. Line 232, “E. coli specific antibodies were modified on the MNPs surface”: what do the authors mean that the antibodies are modified?

5. It would be appropriate to show in the supplementary materials a UV-visible spectrum of AuNPs and MNPs after the conjugation to assess a correct functionalization of the particles.

6. What the spectra of Figure 4a and 4b and Figure 5 represent? Are they the SERS spectra of 4ATP, CTAB or other? A spectral assignment of the peaks, especially of the peak employed for the calibration is required.

7. In Figure 4a and 4b, please report the color legend corresponding to the different spectra reported in the graphs.

8. Line 308, what do the authors mean by “retracting” the blank?

9. Line 314, the LOD calculated according to Figure 6, in which the x-axis corresponds to the logarithm of the E. Coli population, is 107 cfu/mL, not 7 cfu/mL. Please, correct.

10.   Which is the corresponding Limit of Quantification?

11. Regarding the application of the method to milk samples, which are the effect of milk on the SERS spectra? A control SERS spectrum of the AuNRs in presence of milk should be reported.

12. The LOD of the proposed methodology is of 107 cfu/mL, despite this the experiments reported on page 11 employ E.Coli population densities of 120 and 1200 cfu/mL. How authors can be sure that the detection is effective if these values are well below the LOD? Moreover, the corresponding SERS spectra should be reported.

13.   There are several works in the literature that employ SERS for the detection of bacteria (10.1021/acs.analchem.7b02653; 10.1016/j.talanta.2015.09.006; 10.1039/C0AN00473A just to mention a few). In these works, the reported LOD is well below the one of 107 cfu/mL obtained by authors. So, which are the advantages of the methodology proposed by authors with respect to other works that seems to provide more effective results? This aspect should be discussed in the manuscript.

Author Response

Referee 3:

Comment 1:  A careful proofreading of the manuscript is required.

Answer 1: We checked the entire manuscript as the Referee requested.

Comment 2:  In the Introduction, a brief description of SERS spectroscopy and of its applications in bacteria detection would be appropriate in regard to the broad readership of the journal. See for example: 10.1039/d1nr00708d; 10.1039/d0nr06340a; 10.1007/s00604-021-04885-z.

Answer 2: Following phrase and references were added in the Introduction part according to Referee’s suggestion.

“SERS technique made the detection of many molecules having weak Raman signals possible. This sensitivity enhancement made Raman spectroscopy a commonly used detection technique. The enhancement phenomenon lies in the application of nanostructured metal such as silver or gold. Since these metal nanoparticles have enriched optical properties, target molecules were detective in lower detection limits. In addition, the aforementioned nanoparticles have been preferred for labeling in detections and have enabled the multiplex analysis. Nanoparticles are also promising in designing of sensor devices, such as SERS substrates, which are ultimate tools for sensitive pathogen detection.”

Comment 3:  It is well known that gold nanoparticles have also antibacterial properties, see for example: 10.2147/IJN.S134526; 10.1016/j.jcis.2020.07.006; 10.3390/nano11061621; 10.1016/j.jiph.2021.10.007. This aspect should be discussed in the manuscript to highlight the further potentialities of the proposed methodology.

Answer 3: I think, the antibacterial properties of gold nanoparticles is  beyond of the present study.

Comment 4:  Line 232, “E. coli specific antibodies were modified on the MNPs surface”: what do the authors mean that the antibodies are modified?

Answer 4: We apologize for the mistake in writing the modification of MNPs surface. Antibodies bounded to MNP surface by benefiting from avidin-biotin interaction which is stated in the revised manuscript.

Comment 5:  It would be appropriate to show in the supplementary materials a UV-visible spectrum of AuNPs and MNPs after the conjugation to assess a correct functionalization of the particles.

Answer 5: In our previous studies, we demonstrated that a red-shift was observed in the UV-visible spectrum of AuNPs and MNPs afterthe antibody conjugation. However, we did not observe any plasmon band shift antibody-bacteria interaction. Because of this reason, we performed TEM measurements to confirm the functionalization of the particles.

Comment 6:  What the spectra of Figure 4a and 4b and Figure 5 represent? Are they the SERS spectra of 4ATP, CTAB or other? A spectral assignment of the peaks, especially of the peak employed for the calibration is required.

Answer 6: In the present study, SERS spectra of 4ATP labelled sandwich complexes (Au NPs-E. coli-MNPs) obtained after the developed procedure was utilized.

Regarding peak assignment, following phrase was added to part 3.3. in the revised manuscript.

“Since 4-ATP has strong affinity to gold nanoparticles, this molecule can adsorb on the surface via Au-S bonds. As revealed from the SERS spectra, two strong peaks at 1080 and 1590 cm-1 can be observed. While the peaks at 1080 and 1590 cm-1 can be assigned to the a1 modes of 4-ATP molecules on metal surface.”

Comment 7:  In Figure 4a and 4b, please report the color legend corresponding to the different spectra reported in the graphs. 

Answer 7:

Fig.4 is now redrawn and replaced in the text.

Comment 8:  Line 308, what do the authors mean by “retracting” the blank?

Answer 8:

In order to construct calibration curve, the most intense peak values were used for the y-axis values. Blank measurement has also a small value which is retracted from the other y-axis values.

Comment 9:  Line 314, the LOD calculated according to Figure 6, in which the x-axis corresponds to the logarithm of the E. Coli population, is 107 cfu/mL, not 7 cfu/mL. Please, correct.

Answer 9:

We evaluated LOD and LOQ in line with the Referee’s suggestion. Limit of detection value calculated as 7 cfu/mL using the equation SLOD= Sbl +3×sbl, where Sbl is the mean of blank measurements, sbl is the standard deviation of blank measurements.

Comment 10:  Which is the corresponding Limit of Quantification?

Answer 10: The limit of quantification was calculated as 3,3xLOD.

Comment 11:  Regarding the application of the method to milk samples, which are the effect of milk on the SERS spectra? A control SERS spectrum of the AuNRs in presence of milk should be reported.

Answer 11: To explain better this issue, the following statement was added to the revised manuscript (section 3.4).

“Since formed sandwich complexes washed in PBS buffer before the measurement chamber, the effect of milk on the SERS spectra is ignored due to the high intensity of 4-ATP spectrum.”

Comment 12:  The LOD of the proposed methodology is of 107 cfu/mL, despite this the experiments reported on page 11 employ E.Coli population densities of 120 and 1200 cfu/mL. How authors can be sure that the detection is effective if these values are well below the LOD? Moreover, the corresponding SERS spectra should be reported.

Answer 12: In the present study, the LOD value is 7 cfu/mL, not 107 cfu/mL. Here, 120 and 1200 cfu/mL are in our dynamic range (101-107).

Comment 13:  There are several works in the literature that employ SERS for the detection of bacteria (10.1021/acs.analchem.7b02653; 10.1016/j.talanta.2015.09.006; 10.1039/C0AN00473A just to mention a few). In these works, the reported LOD is well below the one of 107 cfu/mL obtained by authors. So, which are the advantages of the methodology proposed by authors with respect to other works that seems to provide more effective results? This aspect should be discussed in the manuscript.

Answer 13: We would like to thank referee for this suggestion. The main limitation in batch type sandwich immunoassay is the time consuming washing steps (about 30 minutes). In this work, conjugation of gold nanoparticles to the MNP- E. coli and washing step was performed in the passive chip and the time necessary for washing out excess gold nanorods was decreased to 1-2 minutes. Furthermore, the detection limit is low compared to the bacteria detection methods and sandwich type studies based on the use of microchips in the literature.  The main advantages of the proposed method are: (i) the time necessary for washing step was decreased, (ii) since the need for an external pump was eliminated the system is simple and portable and  (ii)  the consumption of MNP and gold nanorods is minimized.

The latter statement have been added to last part of the Result and discussion.

Round 2

Reviewer 2 Report

In some cases, the authors did not directly clarify the issues I raised or did not introduce corrections. The following is a summary of my previous comments, the reviewers' responses, and my current comments on the changes made and the clarifications provided.

4) I wonder why the examination was performed only on a single E.coli strain? The fundamental concept of these kind o sensors is based on antibody-modified MNPs which are responsible for the capture of the bacteria cells. However, it is known that because of the genetic similarity among serovars/strains, antibodies or nucleic acid probes show cross-reactions that limit the identification of E.coli serovars. On what basis then do the authors conclude that the developed sensor will allow the detection of different strains of E. coli? The presented results do not demonstrate or confirm this. They only show that a given (more precisely, one) strain has been classified as E.coli. Why did the authors not performed the validation of the proposed method with more than one strain of E.coli?

Answer: The referee is right that the proposed method is specific to E. coli. We indicated serotype of E.coli (ATCC 35218) in section 2.4. The following statement and the related reference was also added to the first paragraph of the section 2.6 to explain better this issue.

“This biotinylated antibody recognizes all ‘O’ and ‘K’ antigenic serotypes of E. coli (https://www.fitzgerald-fii.com/e-coli-antibody-biotin-60-e13b.html (accessed 11 July 2019).”   

Comment: The authors performed experiments only on one E.coli stain, therefore, based on which foundations they conclude that the proposed method will be specific for E.coli species/ all E.coli strains, if they did not provide any results from the examination of different E.coli strains than ATCC 35218?.

5) Why did the authors not decide to prepare the mixture of different bacteria for examination by the proposed method, but analyzed samples of different bacteria separately? The mixture of bacteria is closest to the real microbiological samples? The contamination of the samples by bacteria monoculture is not so common.

Answer: In the present study, in first part of the experiments non-E. coli cultures was used separately at cell densities of 1×104 cfu/mL and the SERS signal was compared with blank signal.  We carried out  experiments with 104 cfu/mL E. coli and SERS signals were compared with 104 cfu/mL E. coli signal.

Comment: One more time they described the methodology of the preformed examination, however, my question was concerning other issue. Most of the biological samples contain mixtures of bacteria and not single species, strains. Therefore, validation of the proposed method must be performed on samples as closely resembling real samples as possible, and in this case, the authors have failed to demonstrate the sensitivity of the proposed biosensor on ‘real’ samples.

11) Line 203-205: “The calibration curve was constructed by using the average of five parallel readings of SERS signal intensity versus logarithmic E. coli concentrations.”- why so few measurements? This allowed for the elimination of noise and proper averaging at all? What measure of uncertainty was used to determine the mean Raman intensity value for a given concentration of bacteria? The standard deviation for 5 measurements is not very meaningful. 

Answer: In section 2.10 sample preparation was described in order to evaluate the applicability of the method to milk samples. The results were presented in Table 1 and discussed in section 3.4.  However, in section 2.9 defines the range for E. coli population density studied in order to determine the linear range for calibration. The population density starting from 1 x 107 cfu/mL and proper dilutions were performed till 1x101 cfu/mL.

In the revision, we provide more detail concerning material and methods. The related statement in section 2.10 was corrected as:

“Three repetitions were performed for each experiment.  The average of all readings (in total 15 for each concentration) was used.”

Comment: I don’t understand, it was explained previously that 3 replicates of the bacteria concentrations were prepared, and now that 3 repetitions of the measurements were performed. So, how did you perform the averaging based on 15 measurements for each concentration?

Moreover, the averaging of the Raman spectra based on only 3 measurements

12) Fig.4: not specifying which colour corresponds to which concentration of bacteria. In recent form, these spectra are not readable. 

Answer: Fig.4 is now redrawn and replaced in the text.

Comment: Still, in the figure caption or in text, there is no explanation of what the different colours of the curves on the Fig.4, Fig.5: refer to.

13) It is possible to obtain the negative value of the Raman intensity (as in Fig.4 and 5) or any other intensity at all? Probably, as indicated in lines 296-297: ‘Each data point is the average of three parallel measurements’ this was due to an insufficient number of repeat measurements, which did not allow effective averaging of the results and elimination of noise? Why did the authors not perform more repeated measurements? Did you extract the background/base line? Why did the authors choose to use the maximum Raman intensity as a measure of bacterial concentration rather than the ratio of the maximum intensities of the two bands?

Answer The background/base line was extracted due to the removal of negative signals. Three repetitions were performed for each experiment.  The average of all readings (in total 9 for each data point) was used. Both approaches can be used to get results as the referee has mentioned.

The referee is right that the ratio of the maximum intensities of the two bands can be also used. Since our calibration curve was linear and signals shows similar trends for the most intense two bands, we preferred to use the maximum Raman intensity.

Comment: The variation of the Raman spectra and negative values of the intensity, which are physically unreasonable to be presented, indicate that the number of measurements repetitions and averaging were not sufficient for elimination of the noise in the signal.

14)Lines 243-244: “TEM image of Au NRs indicates the aspect ratio is about 3 and homogeneous distribution of Au NRs was also observed.”- please explain.

Answer: To explain better, a phrase was added to the part 3.2.

“The rod-shaped gold nanoparticles have an average diameter of 15 nm and an average length of about 45 nm (corresponding aspect ratio of 3).”

Comment: Any information how this evaluation was performed? Manually, based on the REM images or automatically based on some image processing algorithm/plugin? Please provide an explanation in text. Moreover, the dimensions of the rod-shaped nanoparticles will be described better in terms of longitudinal dimension/ width, or length of the major-/ minor-axis rather than diameter and length. Provide the standard deviations related to the determined average quantities.

15)  Fig.6/Fig.7: how were the measurement uncertainties of the individual points on this graph determined? What measure has been used here? How many measurements were made? How many repetitions of bacteria samples were examined? 

Answer: Regarding Fig.6/Fig.7, It was directly calculated by excel program by calculating RSD values of maximum Raman signal intensity of all readings in total 15 for each data point.

Comment: As you wrote previously, the examination was performed based on the 3 replicants of each concentration and 3 repetitions of the Raman spectra measurements, so how the RSD of a single point on Fig.6/Fig.7 can be obtained from 15 measurements for each concentration? It is really confusing. Please

18)  Fig. 2: The following stages of the test, particularly their sequence, are not very well visualized in the diagram in Fig.2.

Answer 18: Fig. 2 was revised according to the Referee’s suggestion

Comment: It was not revised at all. It is the same figures as in the original manuscript submitted.

Additional comments:

1)  Fig.4: The authors are asked to add the error bars to the dependence of Raman intensity on the amount of AuNRS, MNPs.

2) According to Table 1, it is confusing and unreadable. It contains only information about the used (added) and found concentration of bacteria [CFU/ml], but to what is the second column refereeing to: the determined averaged concentration based on the proposed method and its std or the difference between the proposed method and plate counting? Did you initially evaluate the CFU/ml concentration of the samples and then examined them by two methods, or did you used samples with unknown concentration and determined the concentration based on the plate counting and the proposed method? It is not clearly explained in the manuscript. The authors wrote: “Known population densities of E. coli at 120 and 1200 cfu / ml were added to milk samples and analyzed for E. coli”, but these values were obtained by  the plate count?

Moreover, these procedures were not described in detail in the material and methods section.

3) In table 1, two E. coli concentrations 120 and 12000 CFU/ml were examined and not 120 and 1200 CFU/ml as it was stated in lines 385-386: “Known population densities of E. coli at 120 and 1200 cfu / ml were added to milk samples and analyzed …. “.i

4) There is no information on whether plate counting was performed manually or automatically. The authors are asked to provide sufficient information in manuscript.

5) Furthermore, in this examination, the results are shown in Table 1, there is no explanation why such two concentrations (1,2*10^2 and 12*10^3 CFU/ml) were chosen. Why did you not performed examination in range 10^1 and 10^2 CFU/ml near the minimum of Dynamic Range ?

6) How was the LOD equal to 7 CFU / ml  verified  in this study? It was evaluated based on the calibration curve and it is the theoretical LOD rather than experimentally verified value, since in the validation as indicated above the authors decided to used only such 1.2*10^2 and 12*10^3 CFU/ml bacteria concentrations, that are several orders of magnitude greater than the theoretical one, assumed by the authors. The authors are asked to modify their claim or provide additional results that can better support their claim.

Author Response

Comment: The authors performed experiments only on one E.coli stain, therefore, based on which foundations they conclude that the proposed method will be specific for E.coli species/ all E.coli strains, if they did not provide any results from the examination of different E.coli strains than ATCC 35218?.

Answer: Thank you for the referee's valuable comment. Within the scope of the study, a capillary driven microfluidic chip was developed and a new analytical system was developed for the determination of bacteria using the SERS label. E. coli was selected as the model bacteria and quantitative analysis for the determination of bacteria was successfully performed. In our study, polyclonal antibodies were used. Specific serovar of E coli is not targeted. The selectivity of the analysis was ensured by the selectivity of the antibody, and it was also shown that the analysis system did not interact with other bacteria (S. enteritidis and S. aureus) in the environment, except for E coli. Because polyclonal antibody is used, it will produce results in the presence of more than one serotype E coli (antibody recognizes all ‘O’ and ‘K’ antigenic serotypes of E. coli). In the case of using monoclonal antibodies, an analysis system can be developed for specific bacteria determination.

Comment: One more time they described the methodology of the preformed examination, however, my question was concerning other issue. Most of the biological samples contain mixtures of bacteria and not single species, strains. Therefore, validation of the proposed method must be performed on samples as closely resembling real samples as possible, and in this case, the authors have failed to demonstrate the sensitivity of the proposed biosensor on ‘real’ samples.

Answer: Additional experiments were performed and the E. coli signal was recorded using standard addition method as suggested. The following paragraph was replaced with the old one and the results were presented in the revised manuscript in Fig.7, the text and figure legend revised accordingly (Page 11).

“The selectivity of the method for E. coli was evaluated with 1×104 cfu/mL of E. aerogenes and S. enteritidis. Firstly, MNP conjugated to the E. coli selective antibody was interacted with those bacteria in separate vial.  Secondly, same experiment was repeated in the presence of 1×104 cfu/mL E. coli. SERS signals obtained from 1×104 cfu/mL of S. Enteritidis, S. aureus were compared with blank signal and spiked samples were compared with 1×104 cfu/mL E.coli standard signal. Three parallel microchip was used for each experiment. As shown in Fig. 7, the SERS signal remained unchanged, same as blank signal, for S. Enteritidis, S. aureus and the SERS signal intensities of S. Enteritidis, S. aureus solutions (1×104 cfu/mL) both fortified with 1×104 cfu/mL E. coli were the same with the standard containing only 1×104 cfu/mL E. coli (Fig. 7).”

Comment: I don’t understand, it was explained previously that 3 replicates of the bacteria concentrations were prepared, and now that 3 repetitions of the measurements were performed. So, how did you perform the averaging based on 15 measurements for each concentration?

Moreover, the averaging of the Raman spectra based on only 3 measurements

Answer: For AuNRs and MNPs optimization studies (Figure 4 a and b); Three repetitions were performed for each experiment by taking three measurements (in total 9 for each data point).  The average of all readings (in total 15 for each concentration) was used.

For calibration curve studies (Figure 5 and 6);

Three repetitions were performed for each experiment by taking five measurements (in total 15 for each data point).  The average of all readings (in total 15 for each concentration) was used.

Comment: Still, in the figure caption or in text, there is no explanation of what the different colours of the curves on the Fig.4, Fig.5: refer to.

Answer: Fig. 4 has been changed as suggested.

Comment: The variation of the Raman spectra and negative values of the intensity, which are physically unreasonable to be presented, indicate that the number of measurements repetitions and averaging were not sufficient for elimination of the noise in the signal.

Answer: The background/base line may be including low negative signals due to baseline correction. Since we used the most intense peak in the spectrum and all measurements equally influenced by baseline correction, low negative signals in the background affect the most intense peak negligibly.

Comment: Any information how this evaluation was performed? Manually, based on the TEM images or automatically based on some image processing algorithm/plugin? Please provide an explanation in text. Moreover, the dimensions of the rod-shaped nanoparticles will be described better in terms of longitudinal dimension/ width, or length of the major-/ minor-axis rather than diameter and length. Provide the standard deviations related to the determined average quantities.

Answer: To explain better this issue, we added the following,

“The aspect ratio of gold nanoparticles is measured manually based on the TEM images.”

The dimensions of the rod-shaped nanoparticles has been described better in terms of longitudinal dimension/ width, or length of the major-/ minor-axis as the referee has requested. We also provide the standard deviations.

“The length of the major axis of gold nanorod particles is 45±3 nm and the length of the minor axis is 15±3 nm (corresponding aspect ratio of 3).

Comment: As you wrote previously, the examination was performed based on the 3 replicants of each concentration and 3 repetitions of the Raman spectra measurements, so how the RSD of a single point on Fig.6/Fig.7 can be obtained from 15 measurements for each concentration? It is really confusing. Please 

Answer: The 3 replicants of each concentration and 3 repetitions is true for AuNRs and MNPs optimization studies. For calibration studies; the 3 replicants of each concentration and 5 repetitions 3x5=15.

18)  Fig. 2: The following stages of the test, particularly their sequence, are not very well visualized in the diagram in Fig.2.

Comment: It was not revised at all. It is the same figures as in the original manuscript submitted. 

Answer: Fig. 1 has been revised according to the Referee’s suggestion.

Additional comments:

1)  Fig.4: The authors are asked to add the error bars to the dependence of Raman intensity on the amount of AuNRS, MNPs. 

Answer: Fig. 4 is redrawn by adding the error bars.

2) According to Table 1, it is confusing and unreadable. It contains only information about the used (added) and found concentration of bacteria [CFU/ml], but to what is the second column refereeing to: the determined averaged concentration based on the proposed method and its std or the difference between the proposed method and plate counting? Did you initially evaluate the CFU/ml concentration of the samples and then examined them by two methods, or did you used samples with unknown concentration and determined the concentration based on the plate counting and the proposed method? It is not clearly explained in the manuscript. The authors wrote: “Known population densities of E. coli at 120 and 1200 cfu / ml were added to milk samples and analyzed for E. coli”, but these values were obtained by  the plate count?     

Moreover, these procedures were not described in detail in the material and methods section.

Answer: The method and results sections related to the real sample experiment have been rearranged to be more understandable.

Experiments were carried out with skim milk to demonstrate the applicability of the developed sensor system for detection of E. coli in real samples. Prior to analysis, the absence of E. coli in the skim milk samples was confirmed by cultural methods. Then, different concentrations of E. coli (1.2 ´ 102 and 1.2 ´104 CFU/ml) were added to skim milk. These spiked samples were analyzed with the developed chip system and the numbers of E. coli in the samples were determined. The recovery rate was calculated for each concentration as the ratio of the amount detected by the biosensor system to the concentration initially spiked.

Spiked E. coli concentration (CFU/ml)

Detected E. coli concentration (CFU/ml)

Recovery (%)

1.2 ´ 102

1.4 ± 0.3 ´ 102

113

1.2 ´ 104

1.5 ± 0.4 ´ 104

120

3) In table 1, two E. coli concentrations 120 and 12000 CFU/ml were examined and not 120 and 1200 CFU/ml as it was stated in lines 385-386: “Known population densities of E. coli at 120 and 1200 cfu / ml were added to milk samples and analyzed …. “.i

Answer: To clear up the confusion, all bacterial concentrations included in the real sample experiments were expressed more regularly both in the text and in the table.

4) There is no information on whether plate counting was performed manually or automatically. The authors are asked to provide sufficient information in manuscript. 

Answer: We added the following explanation in the revised manuscript.

All bacterial suspensions were adjusted using the McFarland standard, and the numbers of bacteria in the samples were also confirmed by conventional plating methods. The enumeration of E. coli was conducted by spreading the proper dilutions of bacteria on Eosin Methylene Blue Agar (EMBA). Bacterial counts were calculated after the incubation at 37 °C for 18-24 h.”

5) Furthermore, in this examination, the results are shown in Table 1, there is no explanation why such two concentrations (1,2*10^2 and 12*10^3 CFU/ml) were chosen. Why did you not performed examination in range 10^1 and 10^2 CFU/ml near the minimum of Dynamic Range ?

Answer: The main purpose of the trials at this stage is to demonstrate that the developed system can also work in real samples where E. coli can easily be found. The amount of spiked bacteria was chosen based on the similar studies in the literature, so that it could simulate relatively low to moderate E. coli contamination in food (Zhang et al. 2015, Guo et al 2015, Guo et al 2016, Diaz-Amaya et al, 2019).

Zhang, Xinai, et al. "Functionalized gold nanorod-based labels for amplified electrochemical immunoassay of E. coli as indicator bacteria relevant to the quality of dairy product." Talanta 132 (2015): 600-605.

Díaz-Amaya, S., Lin, L.K., Deering, A.J. and Stanciu, L.A., 2019. Aptamer-based SERS biosensor for whole cell analytical detection of E. coli O157: H7. Analytica chimica acta1081, pp.146-156.

Guo, Yuna, et al. "Electrochemical immunosensor assay (EIA) for sensitive detection of E. coli O157: H7 with signal amplification on a SG–PEDOT–AuNPs electrode interface." Analyst 140.2 (2015): 551-559.

Guo, Yuna, et al. "Label-free and highly sensitive electrochemical detection of E. coli based on rolling circle amplifications coupled peroxidase-mimicking DNAzyme amplification." Biosensors and Bioelectronics 75 (2016): 315-319.

6) How was the LOD equal to 7 CFU / ml  verified  in this study? It was evaluated based on the calibration curve and it is the theoretical LOD rather than experimentally verified value, since in the validation as indicated above the authors decided to used only such 1.2*10^2 and 12*10^3 CFU/ml bacteria concentrations, that are several orders of magnitude greater than the theoretical one, assumed by the authors. The authors are asked to modify their claim or provide additional results that can better support their claim.

Answer: We added the following explanation in the revised manuscript.

The limit of detection was calculated as 7 cfu/mL, using the equation SLOD= Sbl+3×sbl, where Sbl is the mean of 10 cfu/mL E. colimeasurements, sbl is the standard deviation of 10 cfu/mL E. coli measurements. The minimum detectable signal (SLOD), calculated from the equation was converted to population density by using a calibration curve constructed with a series of standards.”

Reviewer 3 Report

Authors did not provide satisfactory replies to all the raised issues; therefore, the manuscript did not gain sufficient quality for publication.

The issues not solved are listed below.

 Comment 1: the manuscript is still difficult to understand in several points.

 Comment 5: how can TEM assess the correct functionalization of the AuNPs and MNPs? Moreover the authors must report a comparison between functionalized and non-functionalized AuNPs and MNPs and show that there is a clear difference in the two cases.

 Comment 7: Authors did not report the color legend corresponding to the spectra in In Figure 4a and 4b, as requested.

 Comment 8: Perhaps the authors mean “subtracted”?

 Comment 9: How do authors evaluate the mean of blank measurements and the standard deviation of blank? The values employed for the calculation of the LOD must be reported in the manuscript.

 Comment 11: To ascertain that the effect of milk is negligible, authors must report the corresponding spectra.

Author Response

Referee 3

 Comment 1: the manuscript is still difficult to understand in several points.

Answer: We have substantially revised the manuscript in this respect. Specifically, we would like to point out that experimental part has been reorganized according to the referee suggestions.

 Comment 5: how can TEM assess the correct functionalization of the AuNPs and MNPs? Moreover the authors must report a comparison between functionalized and non-functionalized AuNPs and MNPs and show that there is a clear difference in the two cases.

Answer: In the present study, the gold nanorod particles are coated first with a layer of SERS tag molecules 4-ATP and then a layer of antibodies. The SERS tag molecule is selected to chemisorb as a thiolate adlayer on the gold nanorod particles to provide a strong and unique spectral signature, and to covalently link a layer of antibodies to the gold nanorod particles. As shown in Fig. 5, the SERS spectrum of antibody functionalized 4-ATP-coated gold nanorod samples indicated discrete vibrational peaks at 1590 and 1080 cm-1.

 Comment 7: Authors did not report the color legend corresponding to the spectra in In Figure 4a and 4b, as requested.

Answer: Fig. 4 has been changed as suggested.

 Comment 8: Perhaps the authors mean “subtracted”?

Answer: The term “retracting” has now been changed as “subtracted” as the referee has requested.

 Comment 9: How do authors evaluate the mean of blank measurements and the standard deviation of blank? The values employed for the calculation of the LOD must be reported in the manuscript.

Answer: The limit of detection was calculated as 7 cfu/mL, using the equation SLOD= Sbl+3×sbl, where Sbl is the mean of 10 cfu/mL E. coli measurements, sbl is the standard deviation of 10 cfu/mL E. coli measurements. The minimum detectable signal (SLOD), calculated from the equation was converted to population density by using a calibration curve constructed with a series of standards.

 Comment 11: To ascertain that the effect of milk is negligible, authors must report the corresponding spectra.

Answer: The present work demonstrated that a very small amount of magnetic nanoparticles is spiked directly into the milk sample and selective detection of E.coli is enabled by the use of antibody binding on the capture probe and SERS tag. In this method a gold–thiol monolayer was deposited onto the gold surface and then avidin was immobilized over the SAM surface via covalent binding. In the last step of the surface preparation method, the biotin labeled antibody was adsorbed via avidin–biotin interactions. The rational surface architecture of the antibody via avidin–biotin binding also contributes in an essential way to achieve a high sensitivity as well as low non-specific binding in the milk sample matrix.

Round 3

Reviewer 2 Report

I am grateful for the consideration some of my comments and partial additions to the experimental part of the manuscript. However, despite these corrections, the authors still have not avoided further errors and the manuscript still has some errors which should be corrected:

It was stated that "“The selectivity of the method for E. coli was evaluated with 1×104 cfu/mL of E. aerogenes and S. enteritidis.", but the Authors did not provide any results of this examination. Moreover, the detailed information about the procedure of this examination and  E. aerogenes species (ATTC etc. ) were also not provided in Materials and Methods section. Authors are asked to

Fig. 7: Correct the description of the third bar. Probably it should be "E. coli + S. aureus + S. Enteritidis" and not "E.coli+3S.aureus+E.Enteritids".

According to the " LOD equal to 7 CFU / ml ". As I wrote previously, this value was determined based on the calibration curve and this is rather theoretical than experimentally verified LOD.   Authors are once again asked to indicate in section 3.4 or in Table 2, that this is the theoretical LOD value. Moreover, they are asked to modified the statement  (lines:236-240) as follows:

The theoretical limit of detection was calculated as 7 cfu/mL, using the equation SLOD= Sbl+3×sbl, where Sbl is the mean of 10 cfu/mL E. coli measurements, sbl is the standard deviation of 10 cfu/mL E. coli measurements. The minimum detectable signal (SLOD), calculated from the equation was converted to population density by using a calibration curve constructed with a series of standards.”

Please, correct the names of bacteria, because the nomenclature used is inconsistent and incorrect, for example, bacterial species should be written in italics everywhere in the same manner. Moreover, we do not write S. enteritidis, but S. Enteritidis.

Author Response

Comment: It was stated that "“The selectivity of the method for E. coli was evaluated with 1×104 cfu/mL of E. aerogenes and S. enteritidis.", but the Authors did not provide any results of this examination. Moreover, the detailed information about the procedure of this examination and  E. aerogenes species (ATTC etc. ) were also not provided in Materials and Methods section. Authors are asked to 

Fig. 7: Correct the description of the third bar. Probably it should be "E. coli + S. aureus + S. Enteritidis" and not "E.coli+3S.aureus+E.Enteritids".

Answer: We apologize for the mistake that we made in writing name of bacteria regarding selectivity experiments. It was stated that "The selectivity of the method for E. coli was evaluated with 1×104 cfu/mL of S. aureus and S. Enteritidis."

To confirm the selectivity of the proposed method, Raman signal intensities were measured for Blank, S. aureus,  S. Enteritidis, E. coli, and S. aureus, S. Enteritidis spiked with 1 × 104 cfu/mL E. coli as shown in Figure 7.

Figure 7. SERS signal intensities measured for Blank, S. aureus,  S. Enteritidis, E. coli, and S. aureus, S. Enteritidis spiked with 1 × 104cfu/mL E. coli, respectively.

Comment: According to the " LOD equal to 7 CFU / ml ". As I wrote previously, this value was determined based on the calibration curve and this is rather theoretical than experimentally verified LOD.   Authors are once again asked to indicate in section 3.4 or in Table 2, that this is the theoretical LOD value. Moreover, they are asked to modified the statement  (lines:236-240) as follows:

“The theoretical limit of detection was calculated as 7 cfu/mL, using the equation SLOD= Sbl+3×sbl, where Sbl is the mean of 10 cfu/mL E. coli measurements, sbl is the standard deviation of 10 cfu/mL E. coli measurements. The minimum detectable signal (SLOD), calculated from the equation was converted to population density by using a calibration curve constructed with a series of standards.”

 Answer: We now modified the statement  (lines:236-240) in line with the referee’s suggestion as follows:

“The theoretical limit of detection was calculated as 7 cfu/mL, using the equation SLOD= Sbl+3×sbl, where Sbl is the mean of 10 cfu/mL E. coli measurements, sbl is the standard deviation of 10 cfu/mL E. coli measurements. The minimum detectable signal (SLOD), calculated from the equation was converted to population density by using a calibration curve constructed with a series of standards.”

Comment: Please, correct the names of bacteria, because the nomenclature used is inconsistent and incorrect, for example, bacterial species should be written in italics everywhere in the same manner. Moreover, we do not write S. enteritidis, but S. Enteritidis.

Answer: We again apologize for the mistake regarding the names of bacteria. We now corrected the names of bacteria in the revised manuscript.

Reviewer 3 Report

The authors answered properly to the raiseid issues

Author Response

Comment: Moderate English changes required.

Answer: We have substantially revised the manuscript in this respect.